# ENERGY-BASED TRANSFER FOR REINFORCEMENT LEARNING

## ABSTRACT

Reinforcement learning algorithms often suffer from poor sample efficiency, making them challenging to apply in multi-task or continual learning settings. Efficiency can be improved by transferring knowledge from a previously trained teacher policy to guide exploration in new but related tasks. However, if the new task sufficiently differs from the teacher's training task, the transferred guidance may be sub-optimal and bias exploration toward low-reward behaviors. We propose an energy-based transfer learning method that uses out-of-distribution detection to selectively issue guidance, enabling the teacher to intervene only in states within its training distribution. We theoretically show that energy scores reflect the teacher's state-visitation density and empirically demonstrate improved sample efficiency and performance across both single-task and multi-task settings.

## 1 INTRODUCTION

Reinforcement learning (RL) excels at sequential decision-making (Sutton et al., 1998), but credit assignment, sparse rewards, and modeling errors makes it notoriously sample inefficient. This is limiting in multi-task or continual learning settings, where agents must repeatedly learn to solve new tasks, particularly when those tasks are related to ones they have seen before. A natural question arises: *can we transfer knowledge from previously solved tasks to accelerate learning in new ones*?

One common approach is to reuse a pretrained teacher to guide a student, either directly by suggesting actions (Uchendu et al., 2023) or indirectly by shaping rewards (Brys et al., 2015). This form of transfer learning can be effective: early guidance steers the student toward high-reward behaviors and reduces the need for random exploration. However, when tasks are sufficiently different this approach can impair the student's ability to learn; the teacher may issue sub-optimal guidance that biases exploration towards low-reward regions of the state-action space (Taylor & Stone, 2009).

**In this paper, we introduce an introspective transfer learning method that selectively guides exploration only when the teacher's knowledge is likely to be helpful.** Our approach, *energy-based transfer learning* (EBTL), is based on the insight that guidance should only be issued when the student visits states that lie within the teacher's training distribution. Leveraging concepts from energy-based learning (LeCun et al., 2006) and out-of-distribution detection (Liu et al., 2020), the teacher computes energy scores over states visited by the student during training, treating high-energy states as in-distribution and therefore eligible for guidance. This mechanism enables the teacher to act only when it is sufficiently "familiar" with the current context, making training more efficient not by issuing *more* guidance but by issuing *correct* guidance.

Our contributions are as follows:

- We introduce an energy-based transfer learning method that selectively guides exploration only when the student's state lies within the teacher's training distribution.

- We provide theoretical justification for our approach, showing that the energy score is proportional to the state visitation density induced by the teacher policy.

- We empirically demonstrate that our method yields more sample efficient learning and higher returns than standard reinforcement learning and transfer learning baselines, across both single-task and multi-task settings.

Figure 1: Overview of **energy-based transfer learning**. As the student interacts with the environment, the teacher: 1) checks if each state is in-distribution or out-of-distribution by comparing the state's energy score to a pre-defined *energy threshold*; 2) If the state exceeds the *energy threshold*, then it is considered in-distribution for the teacher and an expert action is suggested to the student.

## 2 RELATED WORK

Reinforcement learning provides a general framework for sequential decision-making, where an agent interacts with an environment to learn a policy that maximizes cumulative reward (Sutton et al., 1998). However, RL methods typically require large amounts of experience, making them sample-inefficient, especially in sparse-reward environments (Andrychowicz et al., 2017). To improve sample efficiency, transfer learning reuses prior knowledge to speed up learning on new tasks (Weiss et al., 2016). A key distinction is whether the teacher interacts with the target task during transfer, yielding *offline* vs. *online* RL transfer.

Offline methods seek robustness by training teacher policies without interaction with the target task, using only source data to learn representations (Bose et al., 2024) or task structure (Rosman & Ramamoorthy, 2012), with the goal of transferring despite covariate shifts. In practice, the absence of target feedback forces reliance either on broad generalization guarantees, which often yield conservative or task-mismatched advice, or on prior assumptions about the target MDP to decide what to transfer, which requires advance domain knowledge. Consequently, offline transfer remains brittle under shift. Beyond fully offline transfer, related strategies reuse source knowledge during training on the target. Pretraining initializes policies, value functions, or representations before online fine-tuning (Abel et al., 2018) and can misguide exploration when source biases are misaligned. Hierarchical transfer employs a high-level controller with options learned in advance (Barreto et al., 2019), but performance degrades when the option library does not cover the target.

By contrast, online transfer learning adapts during the student's training on the target task: a teacher pretrained on a source task monitors rollouts and provides guidance as the student learns. Guidance is delivered interactively, e.g., via action suggestions (Torrey & Taylor, 2013) or reward shaping (Ng et al., 1999). Behavior-based approaches further encourage the student to align with the teacher: policy distillation introduces an auxiliary divergence loss to promote imitation during training (Rusu et al., 2015; Schmitt et al., 2018), while action advising lets the teacher intervene with actions during exploration. Transfer learning may also be viewed from the *student* side (Ilhan et al., 2021), where novelty is measured using an auxiliary target–predictor module. However, such methods do not account for the teacher's familiarity with a given state. A persistent challenge is deciding when to advise, as poorly timed interventions can hinder learning (Torrey & Taylor, 2013). *JumpStart RL* restricts advice to a fixed episode prefix (Uchendu et al., 2023), whereas *introspective action advising* triggers interventions based on deviations in expected return (Campbell et al., 2023).

However, prior approaches, both offline and online, rely on heuristics, fixed hyperparameters, or brittle fine-tuning, which limits generalization. We adopt an online transfer setting because it does not require prior domain knowledge of the target and let the teacher decide state by state what knowledge is beneficial to transfer. Our method builds on this principle by applying theoretically-grounded OOD detection to estimate teacher familiarity and selectively issue guidance, improving transfer under covariate shift.

## 3 BACKGROUND

**Reinforcement Learning.** We study a Markov decision process (MDP), defined by the tuple $(\mathcal{S}, \mathcal{A}, P, R, \gamma)$: $\mathcal{S}$ is the state space; $\mathcal{A}$ the action space; $P(\cdot \mid s, a)$ the transition kernel on $\mathcal{S}$; $R : \mathcal{S} \times \mathcal{A} \to \mathbb{R}$ the reward; and $\gamma \in [0, 1)$ the discount factor. At time $t$, the agent in $s_t \in \mathcal{S}$ chooses $a_t \in \mathcal{A}$, transitions to $s_{t+1} \sim P(\cdot \mid s_t, a_t)$, and receives $r_t = R(s_t, a_t)$. The goal is to learn a policy $\pi(a \mid s)$ maximizing the discounted return $\mathbb{E}_\pi[\sum_{t=0}^\infty \gamma^t r_t]$.

**Energy-Based Out-of-Distribution Detection.** In this work, we determine whether a state is in- or out-of-distribution (OOD) for a given policy. A widely-used baseline for OOD detection uses the maximum softmax probability assigned to a predicted label (Hendrycks & Gimpel, 2016). However, softmax scores are not always reliable as neural networks can produce overconfident predictions for OOD states (Nguyen et al., 2015). An alternative approach is to use the *energy* of a state, which is computed from the raw logits of a network and has been shown to better separate in- and out-of-distribution examples (Liu et al., 2020).

Formally, given $\mathbf{x} \in \mathbb{R}^D$ and network logits $f(\mathbf{x}) \in \mathbb{R}^K$ with components $f_1, \ldots, f_K$, we define the free energy; $T > 0$ is a temperature controlling logit sharpness:

$$E(\mathbf{x}; f) = -T \log \sum_{i=1}^K e^{f_i(\mathbf{x})/T}. \tag{1}$$

An input is considered to be OOD if $E(\mathbf{x}; f) > \tau$ for an *energy threshold* $\tau$ and in-distribution (ID) otherwise. The energy threshold is pre-computed over a set of in-distribution data.

## 4 ENERGY-BASED TRANSFER LEARNING

Our goal is to improve the sample efficiency of reinforcement learning, which is particularly important in multi-task settings where the agent must learn to solve many (potentially related) tasks. One way to improve sample efficiency is to leverage a teacher policy trained on a related source task to guide the student in a new target task. However, naively accepting teacher guidance can degrade sample efficiency if the student visits states outside the teacher's training distribution, potentially biasing exploration toward uninformative or low-reward regions of the state-action space.

To address this, we propose a transfer learning framework in which the teacher suggests actions to the student only in states sufficiently close to the teacher's training distribution. We formalize the problem of when to issue guidance as *out-of-distribution detection for reinforcement learning*.

**Problem Formulation.** Let $\pi_T$ and $\pi_S$ denote the teacher and student policies, respectively. We denote a trajectory as $X = \{x_t\}_{t=1}^n$, where each transition $x_t = (s_t, a_t, s_{t+1}, r_t)$ consists of the state $s_t$, action $a_t$, next state $s_{t+1}$, and reward $r_t$. We define a score function $\phi(s; \pi)$, where a state $s$ is considered ID with respect to a policy $\pi$ if $\phi(s) \geq \tau$, for some threshold $\tau \in \mathbb{R}$, and OOD otherwise. The action selection rule is then defined as:

$$a = \begin{cases} a_T \sim \pi_T(\cdot \mid s), & \text{if } \phi(s; \pi_T) \geq \tau, \\ a_S \sim \pi_S(\cdot \mid s), & \text{if } \phi(s; \pi_T) < \tau. \end{cases} \tag{2}$$

Equation 2 restricts teacher intervention to states sufficiently close to those it has seen during training, deferring to the student policy in all other cases.

**Assumption (Teacher reliability).** We assume the teacher policy $\pi_T$ has been trained to convergence and is near–optimal on its own training distribution. Consequently, whenever a state $s$ is in-distribution for the teacher (i.e., was encountered during teacher training) *or is sufficiently similar to such states in task-relevant features*, the teacher knows how to solve it: guidance from $\pi_T$ is correct, and samples $a_T \sim \pi_T(\cdot \mid s)$ are optimal for that state.

### 4.1 ENERGY SCORES AND STATE VISITATION

We draw inspiration from recent work on energy-based out-of-distribution detection (Liu et al., 2020) and define our score function as the negative free energy of a state $s$ under the teacher policy:

$$\phi(s; \pi_T) = -E(s; \pi_T),$$

where $E(s; \pi_T)$ is the free energy computed from the teacher's network. While our derivation assumes discrete logits, continuous actions can be discretized into bins to compute energy from categorical logits. This follows common practice in continuous-control models (Kim et al., 2024; Zitkovich et al., 2023). We refer to $\phi(s; \pi_T)$ as the energy score, which serves as a proxy for how likely the state is to belong to the teacher's training distribution $p(x)$. In on-policy reinforcement learning, training data is generated by rolling out the teacher policy $\pi_T$ to collect experience. As a result, $p(x)$ is implicitly defined by the state-visitation distribution $d_\pi(s)$ of the teacher. Consequently, the free energy $E(s; \pi_T)$ is negatively related with the teacher's familiarity with a state, assigning lower values to frequently visited states and higher values to unfamiliar ones. Following convention (Liu et al., 2020), we set the energy score $\phi$ to the *negative* free energy so that in-distribution states yield higher scores than out-of-distribution states. In the on-policy setting, an update increases $\phi$ on the visited support (Appendix A.1). Throughout, we assume the teacher policy has converged to a fixed point, and we now relate $\phi$ to state visitation.

**Proposition 4.1.** *Let on-policy training converge to $\theta^\star$, yielding a unique stationary visitation density $d_{\pi_{\theta^\star}}(s)$. If a realizable energy model $p_\theta(s)$ is fit to optimality at convergence, then the log of the visitation density is proportional to the score function $\phi(s) = -E(s)$:*

$$\log d_{\pi_{\theta^\star}}(s) \propto \phi(s).$$

*Proof.* Given an energy-based model $f$, the density of a state $s$ is $p(s; f) = \frac{e^{-E(s;f)/T}}{Z}$, where $Z = \int_s e^{-E(s;f)/T}$ is the partition function and $T$ is the temperature (LeCun et al., 2006). Ignoring the normalizing constant $Z$ and taking the logarithm of both sides we obtain:

$$\log p(s) \propto -E(s).$$

In on-policy RL, training data are collected by sampling trajectories under the current policy $\pi$. At convergence, the policy is fixed ($\pi_{\theta^\star}$) and induces the stationary state distribution $d_{\pi_{\theta^\star}}$. Under realizability, the minimum of $\mathrm{KL}(d_{\pi_{\theta^\star}} \| p_\theta)$ is zero, and under optimality the fitted parameter $\theta^\star$ attains this minimum. Hence $p_{\theta^\star} = d_{\pi_{\theta^\star}}$. Substituting this into the equation above, we obtain:

$$\log d_{\pi_{\theta^\star}}(s) \propto -E(s) = \phi(s).$$

$\square$

---

**Algorithm 1** Energy-Based Transfer for Reinforcement Learning

---

**Input:** Teacher policy $\pi_T$, student policy $\pi_S$, energy threshold $\tau$, decay function $\delta$
**while** not done **do**
    Initialize empty batch $B \leftarrow \emptyset$
    **for** $t = 1 \rightarrow H$ **do**
        Sample $p \sim \mathcal{U}(0, 1)$          $\triangleright$ Sample probability of issuing guidance
        $a_t \leftarrow \begin{cases} \pi_T(a \mid s_t) & \text{if } -E(s_t; \pi_T) \geq \tau \text{ and } p < \delta(t) \quad \triangleright \text{If } s_t \text{ is in-distribution} \\ \pi_S(a \mid s_t) & \text{if } -E(s_t; \pi_T) < \tau \quad\quad\quad\quad\quad \triangleright \text{If } s_t \text{ is out-of-distribution} \end{cases}$
        Take action $a_t$, observe $r_t, s_{t+1}$
        $B \leftarrow B \cup (s_t, a_t, s_{t+1}, r_t)$
    **end for**
    Update $\pi_S$ with batch $B$          $\triangleright$ Any on-policy update
**end while**

---

## 4.2 ALGORITHM

Algorithm 1 summarizes EBTL. The student interacts with the environment, while selectively receiving guidance from a teacher policy. At each timestep, EBTL evaluates whether the current state is familiar to the teacher using an energy-based OOD score. If the state is deemed ID and a decaying probability schedule permits guidance, the action is sampled from the teacher policy; otherwise, the student policy acts. To decide when to issue guidance, we compute $\tau$ (see Equation 2) as the empirical $q$-quantile of energy scores over teacher training states $\mathcal{S}_T$, i.e., $\tau = \mathrm{Quantile}_q(\{\phi(s) \mid s \in \mathcal{S}_T\})$. Following prior work (Schmitt et al., 2018; Uchendu et al., 2023; Campbell et al., 2023), we apply a linear decay schedule $\delta(t)$ to control the probability of guidance. This enables early reliance on the teacher while gradually promoting student autonomy. We also incorporate importance-ratio correction, as teacher actions $a_t \sim \pi_T(\cdot \mid s_t)$ are off-policy for the student (Campbell et al., 2023).

**Energy Regularization.** As discussed in Section 4.1, the score function $\phi(s)$ is related to the teacher's state-visitation frequency: frequently visited states tend to receive higher scores. This relation, however, provides no guarantee for states that lie outside the teacher's experience. OOD states may receive heterogeneous scores, some overlapping with in-distribution values, which undermines separability. Moreover, collapsing all out-of-distribution states to a uniform low score would erase meaningful distinctions between states that are structurally compatible with the teacher's support and those that are not. To address this, we incorporate an energy regularizer that enforces a margin between in-distribution and truly out-of-distribution scores, thereby preserving the relative ordering of familiar states while ensuring reliable separation.

To improve separability, we adopt the energy-based loss of (Liu et al., 2020), and augment teacher training with a fixed set of representative OOD states. Let $\mathcal{D}_{\text{in}}^{\text{train}}$ denote the set of ID states collected during teacher training and $\mathcal{D}_{\text{out}}^{\text{train}}$ a curated set of OOD states. Let $\mathbf{s}_{\text{in}} \sim \mathcal{D}_{\text{in}}^{\text{train}}$ and $\mathbf{s}_{\text{out}} \sim \mathcal{D}_{\text{out}}^{\text{train}}$ denote samples from each. Using the energy score $\phi(s) = -E(s)$, the loss is defined as:

$$\mathcal{L}_{\text{energy}} = \mathbb{E}_{\mathbf{s}_{\text{in}}}\left[\left(\max(0, m_{\text{in}} - \phi(\mathbf{s}_{\text{in}}))\right)^2\right] + \mathbb{E}_{\mathbf{s}_{\text{out}}}\left[\left(\max(0, \phi(\mathbf{s}_{\text{out}}) - m_{\text{out}})\right)^2\right],$$

where $m_{\text{in}} \in \mathbb{R}$ and $m_{\text{out}} \in \mathbb{R}$ are margin thresholds for ID and OOD energy scores, respectively. The loss penalizes ID energies $< m_{\text{in}}$ and OOD energies $> m_{\text{out}}$. Appendix E.1 shows separability is insensitive to the choice of $m_{\text{in}}$ and $m_{\text{out}}$. The overall teacher loss is $\mathcal{L}_{\text{total}} = \mathcal{L}_{\text{RL}} + \lambda \cdot \mathcal{L}_{\text{energy}}$ where $\lambda \in \mathbb{R}^+$ controls the weight of the energy regularization. In EBTL, OOD samples are drawn from random policy rollouts in the target environment. ID samples are drawn from the teacher's own training experience via random subsampling. When such data are not available, the ID distribution can be obtained by rolling out the teacher, yielding similar transfer performance (Appendix E.2). We also emphasize that adding the term of regularization of energy does not compromise the teacher's final performance; rather, it accelerates convergence. (Appendix E.3)

## 5 EXPERIMENTS

We evaluate our method in two settings: **single-task transfer** and **multi-task transfer**. The single-task setting is a Minigrid-based (Chevalier-Boisvert et al., 2023) environment composed of navigation tasks, where the agent's objective is simply to reach a goal location. In the multi-task setting, we use Overcooked (Carroll et al., 2019), where the agent must learn to solve multiple task variants, such as how to cook different recipes. For each setting, we design scenarios with increasing covariate shift between the teacher's training distribution and the student's target distribution to test robustness under progressively harder transfer.

In each domain, we examine learning performance with the goal of understanding: (1) whether our method leads to improved sample efficiency, and (2) when the teacher chooses to provide guidance during the student's learning process. We compare our approach against the following baselines:

- **No Transfer:** An agent trained from scratch with standard RL.
- **Action Advising (AA):** A teacher provides advice at every timestep. Advice issue rate decays over time using a predefined schedule.
- **Fine-Tuning:** The student is initialized from a pretrained teacher policy. Convolutional layers are frozen, and only the remaining parameters are updated during training.
- **Kickstarting RL (KSRL)** (Schmitt et al., 2018): A distillation method that applies a decaying cross-entropy loss between student and teacher policies to promote imitation.
- **JumpStart RL (JSRL)** (Uchendu et al., 2023): A time-based method where the teacher advises only early in each episode, using a decaying timestep threshold.

All experiments use teacher and student policies trained with TorchRL's proximal policy optimization (Bou et al., 2023). Full hyperparameter are in Appendix C.

### 5.1 SINGLE-TASK SETTING: MINIGRID

Our Minigrid environment consists of four interconnected rooms and serves as a controlled single-task setting. We design two transfer setups, as illustrated in Figure 2a:

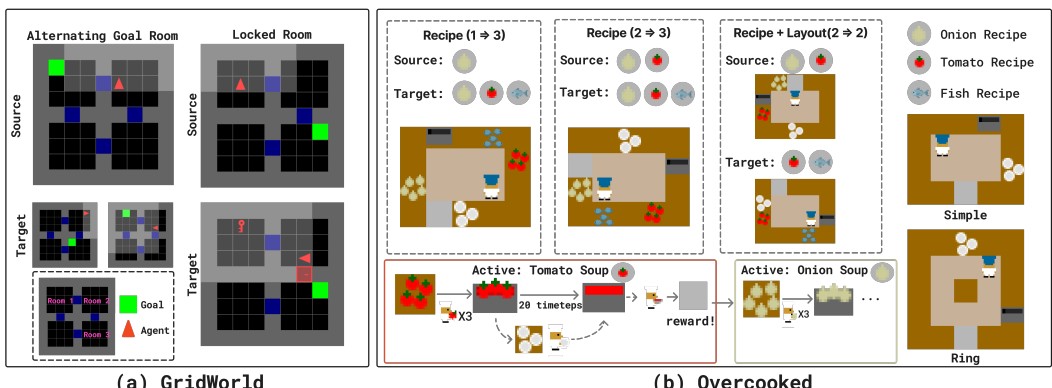

**(a) GridWorld**  **(b) Overcooked**

Figure 2: Environments used for empirical experiments. Refer to Section 5 for detailed descriptions.

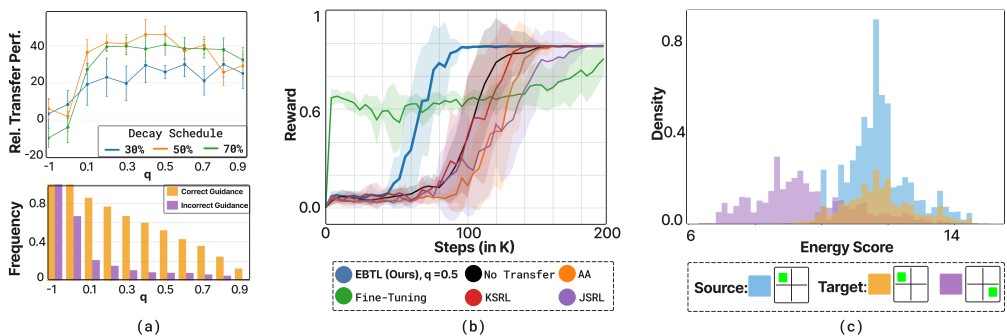

Figure 3: **Alternating Goal** (10 seeds). **(a, Top)** Relative transfer (%) by $q$ and decay schedule; (50% means the guidance probability reaches 0 at mid-training). $q = -1$ advise in all states (AA Baseline). **(a, Bottom)** Fraction of correct vs. incorrect guidance ("correct" = issued on in-distribution states). **(b)** Evaluation returns for EBTL vs. baselines. **(c)** Empirical energy score distributions with respect to the teacher policy. The source task (blue) shows the teacher's training distribution. The target task (orange + purple), measured during transfer, is bimodal: one mode overlaps with the source (shared sub-task, ID), the other does not (non-shared sub-task, OOD).

**Alternating Goal Room.** The source task always places the goal in a random location in Room 1 (upper-left), while the target task randomly places it in either Room 1 (upper-left) or Room 3 (lower-right). The teacher should intervene only when the goal is in Room 1, where its prior experience applies; when the goal is in Room 3, the student must act independently.

**Locked Room.** In the source task, the agent can freely move between rooms. In the target task, a locked door blocks access to the lower area, and the agent must first retrieve a key (randomly placed in the upper rooms) to open it. Because the teacher was never trained with keys, its guidance is useful only after the key is collected, when the remaining navigation matches prior experience. This door–key dependency alters both the encountered states and the path to the goal, creating a significantly larger covariate shift between target and the source task.

The results for the Alternating Goal Room and Locked Room setups are illustrated in Figure 3 and Figure 4, respectively. We make the following observations.

**EBTL consistently outperforms all baselines and is robust to hyperparameters.** In both transfer setups, EBTL achieves the highest sample efficiency of all baselines. For $q \geq 0.1$, EBTL rarely issues guidance in unfamiliar states, leading to significant improvements in transfer performance. As in Fig. 5b, the teacher assigns higher energy to states seen in training (goal in Room 1, upper-left) than to unseen states (goal in Room 3, lower-right). EBTL has two parameters: the energy quantile $q$ and the decay schedule $\delta(t)$. Across varying decay schedules (Figures 3a and 4a), performance is stable: increasing $q$ initially improves transfer by filtering harmful OOD advice, whereas excessively

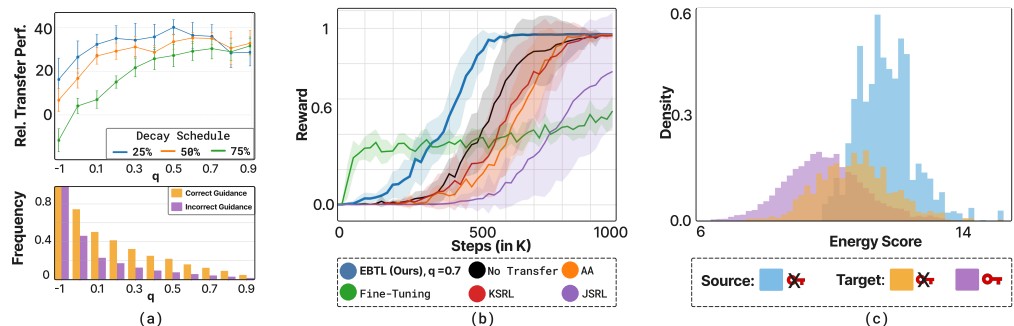

Figure 4: **Locked Room** (10 seeds). See Figure 3 for caption details.

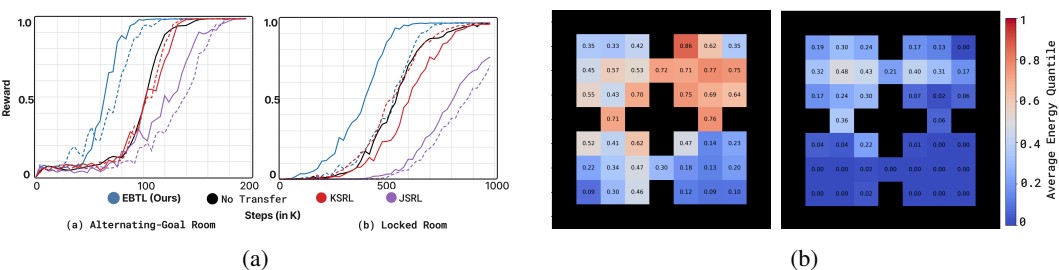

Figure 5: **(a)** Transfer performance with (solid) vs. without (dashed) energy regularization. 10 seeds. **(b)** Heatmaps of average energy quantiles under the teacher for Alternating Goal. Left: source (goal in Room 1). Right: target (goal in Room 3). Higher quantiles indicate greater teacher familiarity.

large $q$ suppresses useful ID guidance. While the optimum is near $q \approx 0.5$, performance is nearly identical for $q \in [0.2, 0.8]$, indicating robustness to the hyperparameter choice.

**Higher covariate shift makes OOD detection more challenging.** In Alternating Goal Room, ID and OOD are well separated (Fig. 3c). In Locked Room, the door–key novelty induces stronger covariate shift, reducing separation (Fig. 4c). Despite this, the teacher still assigns lower energy scores to pre-key states compared to post-key states, indicating meaningful ID/OOD discrimination.

**Energy regularization significantly improves EBTL but has little effect on other methods.** As shown in Figure 5a, adding energy loss speeds up EBTL's convergence, especially in the harder Locked Room with greater covariate shift. In contrast, other baselines show little to no change whether the teacher uses energy regularization. Notably, even without energy loss, EBTL still exceeds all baselines, highlighting the robustness of our approach.

**EBTL improves monotonically as teacher proficiency increases.** Although EBTL assumes a reasonably trained teacher, we ablate robustness under suboptimality. We train teachers at multiple proficiency levels of maximum returns (Table 1). As proficiency rises, advice quality improves and EBTL's transfer gains increase. In contrast, baselines that imitate indiscriminately often fail to benefit from stronger teachers and can deteriorate, because source-task optimality misaligns with the student's environment; in unseen states, copying actions can perform as poorly as a random policy. Energy gating limits advice to states within the teacher's support, mitigating this mismatch.

## 5.2 MULTI-TASK SETTING: OVERCOOKED

We create a single-agent variant of the popular Overcooked (Carroll et al., 2019) environment designed to evaluate multi-task learning. This environment is both *long-horizon* and *high-dimensional*.

**Long-horizon.** At each timestep, exactly one recipe (onion, tomato, or fish soup) is active. Producing a single soup requires the agent to fetch and place three matching ingredients into a pot, wait 20 steps for cooking, retrieve a dish, and deliver the soup to the serving station to obtain reward. After each delivery (correct or not), a new recipe is sampled from the allowed set.

Table 1: **Robustness to suboptimal teachers**. Relative transfer (%) vs. scratch (mean ± 95% CI, 10 seeds) for Alternating Goal. Guidance ends at mid-training.

| Optimality | EBTL (0.3) | EBTL (0.5) | EBTL (0.7) | AA | JSRL | KSRL | Fine-tuning |
|---|---|---|---|---|---|---|---|
| 96% | 41.2±4.9 | 46.1±4.5 | 40.3±4.7 | 6.1±5.6 | −3.2±10.3 | 17.5±6.7 | −32.2±12.9 |
| 70% | 26.8±4.6 | 31.2±5.7 | 27.0±5.4 | 17.0±4.3 | 24.8±5.4 | 14.2±6.1 | −104.3±8.6 |
| 45% | 16.7±4.6 | 14.3±4.3 | 18.9±9.2 | 13.9±6.2 | 15.9±3.7 | 21.5±5.7 | −80.6±3.0 |
| 15% | 9.2±5.9 | 12.8±4.7 | 13.7±4.9 | 12.5±7.1 | 5.0±6.4 | 22.9±4.3 | −110.5±4.4 |

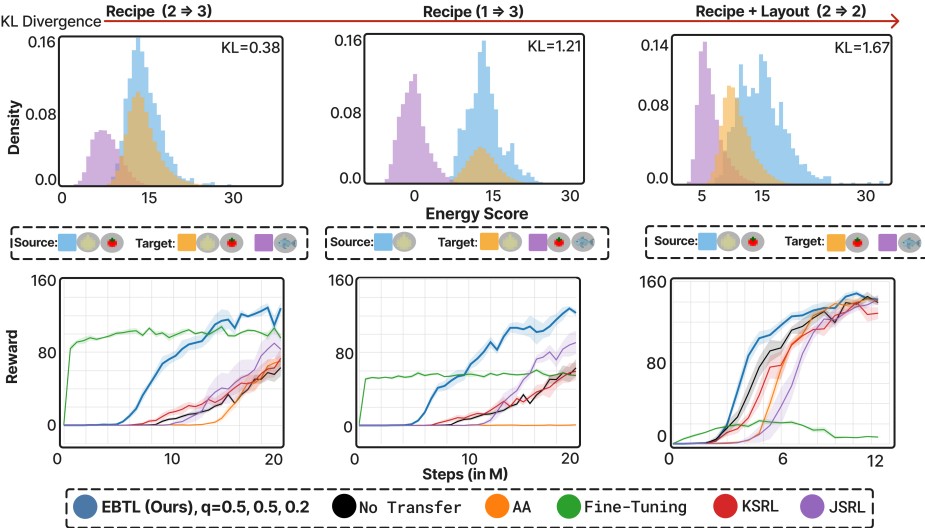

Figure 6: **Simple Room** (5 seeds). **(Top)** Empirical energy score distributions with respect to the teacher policy. The source task (blue) shows the teacher's training distribution. The target task (orange + purple), measured during transfer, is bimodal: one mode overlaps with the source (shared sub-task, ID), while the other does not (non-shared sub-task, OOD). **(Bottom)** Evaluation returns for EBTL and baselines. $q = 0.5$ for Recipe Shift and $q = 0.2$ for the Recipe + Layout Shift.

**High-dimensional.** The state space is combinatorial, due to randomized placement of ingredient dispensers, pots, and serving stations; pot contents and cook-timer values; counter inventories; held objects; agent orientations; and so on. Together these factors yield over $10^{12}$ states even in the simplest layout, making explicit visitation counting infeasible. (Appendix B.2).

An overview of the setup is shown in Figure 2b. We evaluate on two rooms of increasing complexity: a simple room and a ring-shaped room. In this cooking environment, observations encode the current recipe, enabling the agent to differentiate between tasks. Consequently, covariate shift arises along two axes: recipe changes modify what the agent is asked to produce, while layout changes alter the dynamics and the routes available. For each room, we construct three Overcooked transfer setups with increasing levels of covariate shift between the teacher and student environments:

**Recipe Shift (2 ⇒ 3):** Both the source and target environments include all three ingredients: onions, tomatoes, and fish. The source task requires onion and tomato soup, while the target task requires onion, tomato, and fish soup resulting in recipe shift.

**Recipe Shift (1 ⇒ 3):** Both environments again have all three ingredients. This time, the source task requires only onion soup while the target task requires onion, tomato, and fish soup, introducing a higher degree of recipe shift.

**Recipe + Layout Shift (2 ⇒ 2):** The source environment includes only onions and tomatoes and requires onion and tomato soup, while the target environment includes only tomatoes and fish and requires tomato and fish soup, resulting in both recipe and layout shift.

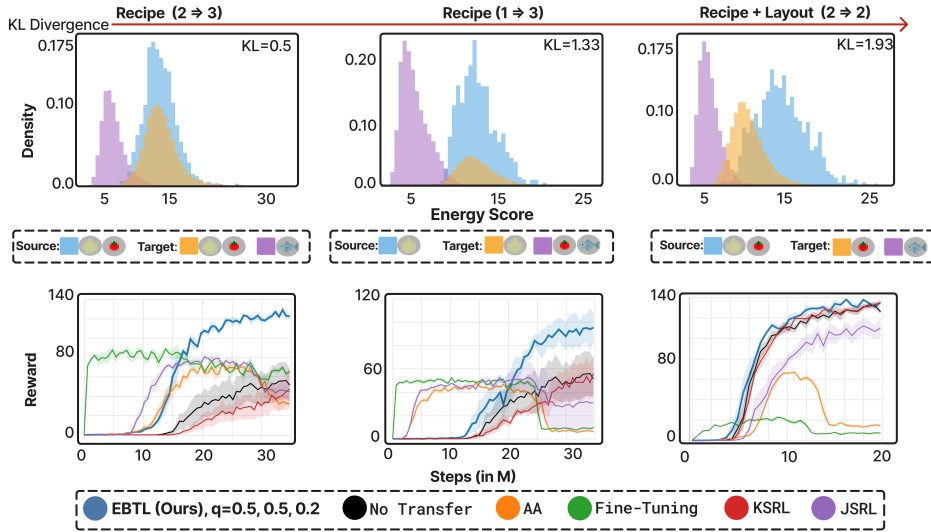

Figure 7: **Ring Room** results (5 seeds). See Figure 6 for caption details.

The results for the Simple Room and Ring Room setups are shown in Figure 6 and Figure 7, respectively. The relative difficulty of each transfer scenario is reflected by the increasing KL divergence between the energy score distributions over source and target states.

**EBTL maintains positive transfer under increasing covariate shift.** EBTL consistently outperforms all baseline methods in both sample efficiency and final policy return across all scenarios. As covariate shift between the source and target environments increases, transfer becomes more challenging. This is evident in the slower convergence from Recipe (2 ⇒ 3) to Recipe (1 ⇒ 3) in the Ring Room (Figure 7), where the teacher is only familiar with 1 rather than 2 recipes (out of 3 total). Despite this, EBTL yields positive transfer performance by restricting guidance to states associated with recipes that the teacher has encountered during training.

**Shared layouts simplify OOD detection.** In scenarios where the source and target tasks share spatial layouts, i.e. Recipe (2 ⇒ 3) and Recipe (1 ⇒ 3), the covariate shift is due entirely to the recipe encoding in the observation. This results in a clearly bimodal energy distribution in the target task (one mode for ID states and another for OOD) simplifying the OOD detection problem (refer to the top row of Figure 6 and Figure 7). However, when the layout changes, as in Recipe + Layout (2 ⇒ 2), there is a systematic decrease in ID energy scores, blurring the ID/OOD boundary. This is because even states associated with familiar recipes appear slightly OOD due to the layout shift.

**Robustness to layout complexity.** EBTL achieves stable positive transfer across both Overcooked environments, resulting in high returns and low variance. In contrast, baselines without selective guidance, such as action advising (AA), degrade as layout complexity increases. As shown in Figure 7, AA performance becomes unstable over training, indicating that over-reliance on suboptimal advice hinders learning. Fine-tuning likewise often stalls in local minima and fails to converge under covariate shift: a misaligned initialization biases early exploration toward low-reward regions, reinforcing bad value estimates and impeding recovery.

## 5.3 ONLINE TRANSFER LEARNING IN CONTINUOUS CONTROL

EBTL also applies to continuous control: the action space can be discretized into bins (as in Open-VLA (Kim et al., 2024)), or an energy score can be defined directly in the continuous domain. Appendix I provides the full continuous formulation (C-EBTL). In this case, the teacher policy outputs a mean and diagonal covariance for each state $s$, and the energy score is derived from the log-determinant of the covariance: $\phi(s) = -E(s) = \frac{1}{2}\log|\Sigma(s)| = \sum_{i=1}^{D}\log\sigma_i(s)$, where $\sigma_i(s)$ is the standard deviation of the $i$-th action dimension. We evaluate on Meta-World(Yu et al., 2020), a suite of continuous-control manipulation tasks for a simulated Sawyer arm. Both state and action spaces are continuous; the action space is four-dimensional, consisting of three Cartesian end-effector displacements plus a gripper command. We consider two settings: window manipulation (Window

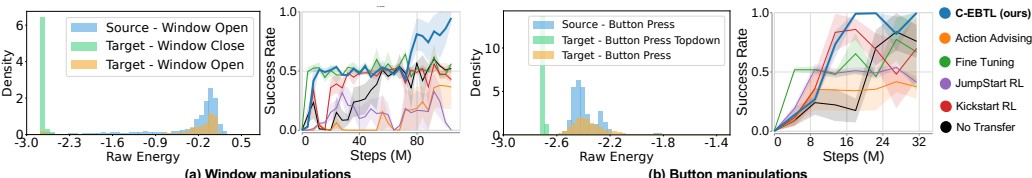

Figure 8: Results (3 seeds). In each environment, the left column shows the empirical density of the teacher's state energy $\phi$. Blue traces the source rollouts used to train the teacher. Orange and green are target rollouts during transfer and typically form a bimodal pattern: one component overlaps the source shared task, in distribution), while the other occupies a separate region (non-shared task, out of distribution). $q$ is set to 0.3 in both settings.

Open, Window Close) and button manipulation (Button Press, Button Press Down). In the window setting the teacher is trained on Window Open; in the button setting the teacher is trained on Button Press. In each setting, the *student must learn both tasks in that pair*.

**Energy score separates ID from OOD.** In both environments the distributions are clearly bimodal: states from the unseen target subtask cluster at a lower mode, while states from the teacher's training subtask occupy a higher one. Across both settings, the OOD mode concentrates near $-2.8$. This follows from the floor we impose on the policy standard deviations for numerical stability in the network, $\sigma_i \geq 0.5$, which implies $\phi(s) = \sum_{i=1}^{4} \log \sigma_i(s) \approx -2.8$. In contrast, the ID energies differ between settings, consistent with Proposition 3.1: when a state admits many compatible actions, on-policy updates raise $\phi_T(s)$. Window manipulation allows more compatible actions (approach the handle, then slide to the target), so its ID energies are higher. Button pressing requires precise alignment and actuation, so fewer actions are compatible and the ID energies remain lower than in window tasks, but still above the OOD mode.

**Consistent robustness in transfer.** As shown in Fig. 8, C-EBTL outperforms all baselines. In Window Manipulation, the learning curve exhibits two clear jumps. The first, around 10M steps, corresponds to rapid acquisition of *window open*. With C-EBTL the student is steered toward teacher-familiar states, so *window open* is learned first. In contrast, the No Transfer baseline lacks this steering and does not reach a comparable level on its first mastered task until about 60M steps. The second jump for C-EBTL, near 70M steps, marks learning *window close*, which the teacher has not seen. Methods without advice

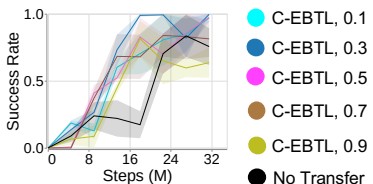

Figure 9: Transfer under different $q$

filtering fail to reach optimal success, because unfiltered suggestions in out-of-distribution states disrupt learning on the unseen subtask. Figure 9 shows that C-EBTL is robust to the choice of $q$. For $q \leq 0.5$, it reaches the optimal success rate at comparable timepoints across settings. When $q > 0.5$, performance declines gradually because a larger $q$ raises $\tau$ and reduces the frequency of advice; with too little advice, the student benefits less from the teacher. Overall, C-EBTL remains strong and stable for reasonable $q$ (e.g., $q \leq 0.5$).

## 6 CONCLUSION AND FUTURE DIRECTIONS

We introduced energy-based transfer learning (EBTL), which improves sample efficiency in reinforcement learning through selective teacher guidance. EBTL employs the teacher's energy as a familiarity proxy, issuing advice only in likely in-distribution states and thereby avoiding additional networks, mappings, or handcrafted OOD detectors. Empirically, EBTL outperforms baselines in both single- and multi-task transfer. A natural extension is to handle multiple teachers in continual learning, selecting at each state the teacher with highest estimated familiarity. A key limitation is that our guarantees rely on the on-policy link between energy and visitation density; under off-policy training this connection weakens, and the theory no longer directly applies.

ETHICS STATEMENT

This work does not raise any specific ethical concerns. Our study does not involve human subjects, sensitive data, or applications with foreseeable harmful impacts. All datasets and environments used are publicly available and widely adopted in prior research. We have taken care to ensure that our methodology is transparent and that our experiments are reproducible. To further support reproducibility, we have included the code used in our experiments, and we are confident that our results can be independently verified by other researchers.

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

APPENDIX

## A  SUPPORTING PROOF

### A.1  ON-POLICY UPDATES INCREASE THE ID SCORE

**Proposition .1** (On-policy monotonicity of the energy score)**.** *Under on-policy training, an on-policy step that maximizes a weighted log-likelihood pushes up the energy score $\phi_\theta(s)$ on visited states.*

*Proof.* Let $\pi_\theta(a \mid s) = \mathrm{softmax}(f_\theta(s)/T)_a$ with logits $f_\theta(s) \in \mathbb{R}^K$ and temperature $T > 0$. An on-policy step maximizes $J(\theta) = \mathbb{E}_{(s,a)\sim d_\pi}[w(s,a) \log \pi_\theta(a \mid s)]$ with weights $w(s,a) \geq 0$. For a single sample $(s,a)$ with weight $w$,

$$\mathcal{L}_{\mathrm{on}}(\theta; s, a) = -w \log \pi_\theta(a \mid s) = w \left[ \frac{1}{T} E_\theta(s,a) + \log \sum_{j=1}^{K} e^{-E_\theta(s,j)/T} \right].$$

Differentiating,

$$\frac{\partial \mathcal{L}_{\mathrm{on}}}{\partial \theta} = \frac{w}{T} \left[ \left(1 - \pi_\theta(a \mid s)\right) \frac{\partial E_\theta(s,a)}{\partial \theta} - \sum_{j \neq a} \pi_\theta(j \mid s) \frac{\partial E_\theta(s,j)}{\partial \theta} \right].$$

Thus a descent step lowers $E_\theta(s,a)$ and raises $E_\theta(s,j)$ for $j \neq a$. Write the free energy as

$$E_\theta(s) \; := \; -T \log \sum_{j=1}^{K} e^{f_\theta(s)_j/T}, \quad \text{so that} \quad \frac{\partial E_\theta(s)}{\partial f_\theta(s)_j} = -\pi_\theta(j \mid s).$$

The update increases $f_\theta(s)_a$ and decreases $f_\theta(s)_j$ for $j \neq a$, hence $E_\theta(s)$ decreases and the score $\phi_\theta(s) = -E_\theta(s)$ increases for this sample. Averaging over nonnegative on-policy weights yields the claim on the visited support. $\square$

## B  TRAINING DETAILS

### B.1  GRIDWORLD

**Reward Structure and Action Masking.**  In the MiniGrid experiments, agents are trained under a sparse reward setting: a reward of 1 is given only when the agent successfully reaches the goal location. No shaped or intermediate rewards are provided, making the task highly exploration-dependent. To mitigate the resulting challenge and accelerate learning, we apply action masking to dynamically restrict the agent's action space based on its immediate environment. The action mask disables irrelevant or invalid actions at each timestep: (1) the *forward* action is masked out if the agent is facing a wall, preventing redundant collisions; (2) the *pickup* action is disabled unless the agent is directly facing a key; (3) the *toggle* action is masked out unless the agent is facing a door; (4) the *drop* action is always disabled, as object dropping is unnecessary in our tasks; and (5) the *done* action is permanently disabled, since it is not used in our environments. This selective pruning of the action space reduces the likelihood of unproductive behavior and enables the agent to focus on learning goal-directed policies more effectively.

**Teacher Training.**  In both experimental setups, we train two variants of the teacher policy using standard Proximal Policy Optimization (PPO) in the source environment: one with the energy-based loss and one without. For the teacher trained with energy loss, the $m_{\mathrm{in}}$ and $m_{\mathrm{out}}$ are set to 10 and 15 respectively. These values are chosen arbitrarily, as the separation between energy distributions is insensitive to the exact threshold choice (see Section E.1). The training follows a consistent set of hyperparameters, as detailed in the next section. For the *unlocked-to-locked* environment, 800K-step checkpoints are selected from both training variants. For the *alternating-goal room* environment, 200K-step checkpoints are used.

**Student Training.** For each target task, we first train a student policy from scratch using standard PPO without any transfer to establish baseline performance. In the *unlocked-to-locked* environment, the total training horizon for transfer experiments is set to 1 million steps, while in the *alternating-goal room* environment, it is set to 200,000 steps. All experiments in the MiniGrid setups are conducted with 10 random seeds to ensure robustness. Within each domain, the student and teacher policies share the same model architecture.

## B.2 OVERCOOKED-AI

**State Space Enumeration of Overcooked** Consider the Simple Layout illustrated in Figure 2. The grid has dimensions $4 \times 5$. The interior region contains 6 traversable tiles where the agent can move. The exterior non-corner tiles are reserved for environment objects (ingredient dispensers, pot, dish dispenser, serving counter), and their placements can be randomized.

We explicitly count states under the *single-agent, lossless* encoding generated by `lossless_state_encoding_single_agent()` in `OvercookedGridWorld`: The resulting state space is a combinator of the following factors.

- **Kitchen-layout permutations.** Among the 14 exterior tiles, 8 are eligible for randomization. We must place 6 *distinct* stations (*onion*, *tomato*, *fish* dispensers; *server*; *pot*; *dish* stack), leaving the remaining 2 tiles as empty counters. The number of distinct assignments is
$$\binom{8}{6} 6! = {}_8P_6 = \frac{8!}{2!} = 20{,}160.$$

- **Agent position and orientation**: The agent may occupy any of the 6 interior tiles and face one of 4 directions (north, south, east, west), for a total of $6 \times 4 = 24$ possibilities.

- **Urgency flag**: Binary indicator with 2 possibilities, set to 1 if the remaining horizon is less than 40, and 0 otherwise.

- **Active recipe**: 3 possibilities, indicating the current recipe type (onion, tomato, or fish).

- **Pot state (mode + contents/recipe/timer)**: Idle with $k \in \{0, 1, 2\}$ ingredients (order irrelevant): $\sum_{k=0}^{2} \binom{k+2}{2} = 1 + 3 + 6 = 10$. With 3 ingredients, the pot is *cooking*: there are $\binom{5}{3} = 10$ recipe multisets, each with a remaining time in $\{1, \dots, 20\}$, giving $10 \times 20 = 200$ states. When cooking finishes, it is *done* with one of the same 10 recipes. Total $= 10 + 200 + 10 = 220$.

- **Agent hand**: The agent may hold (i) nothing, (ii) an empty dish, (iii) a finished soup (all 3 slots filled with a combination of onion, tomato, and fish), or a single raw ingredient (onion, tomato, or fish). The number of distinct soup types is $\binom{3+3-1}{3} = \binom{5}{3} = 10$. Hence the total possibilities are
$$1 \text{ (nothing)} + 1 \text{ (empty dish)} + 10 \text{ (soup types)} + 3 \text{ (single ingredient)} = 15.$$

- **Counter items (2 exterior counters)**: Each counter has 15 options (empty; three ingredients; empty dish; ten soup types), giving $15^2$ overall.

Multiplying the independent factors above gives:
$$|\mathcal{S}| = \underbrace{20{,}160}_{\text{layout}} \times \underbrace{24}_{\text{agent pos/orient}} \times \underbrace{2}_{\text{urgency}} \times \underbrace{3}_{\text{active recipe}} \times \underbrace{220}_{\text{pot state}} \times \underbrace{15}_{\text{agent hand}} \times \underbrace{15^2}_{\text{two counters}}$$
$$= 2{,}155{,}507{,}200{,}000 \approx 2.16 \times 10^{12}.$$

This already conservative count highlights why explicit state-visitation is infeasible for the teacher model; more complex layouts such as the *Ring* further enlarge the state space.

**Reward Structure.** In all Overcooked setups, no action masking is applied. Instead, shaped rewards are introduced to facilitate the training process. A shaped reward of 3 is given when the correct ingredient is added to a pot. An additional reward of 3 is awarded when a dish is picked up—provided there are no dishes already on the counter and the soup is either cooking or completed. A reward of 5 is granted when the soup is picked up. Furthermore, a shaped reward of 3

is given upon delivering the soup, regardless of whether it matches the currently active recipe. All shaped rewards follow a predefined linear decay schedule. In contrast, a sparse reward of 20 is awarded when the delivered soup matches the active recipe; this reward does not decay over time.

**Teacher Training.** In all Overcooked setups, teacher policies are trained in the source environment using standard Proximal Policy Optimization (PPO) with hyperparameters described in the following section. For each setup and source-target configuration, a specific checkpoint is selected to serve as the teacher for transfer. The table below lists the selected training step (in environment steps) corresponding to each teacher checkpoint.

Table 2: Selected teacher checkpoints (in environment steps) for each Overcooked setup and source-target configuration.

| Setup | Recipe ($2 \rightarrow 3$) | Recipe ($1 \rightarrow 3$) | Recipe + Layout ($2 \rightarrow 2$) |
|---|---|---|---|
| Simple Room | 19,008,000 | 9,004,800 | 12,000,000 |
| Ring Room | 2,400,000 | 10,003,200 | 18,000,000 |

**Student Training.** In all Overcooked setups, student policies are trained in the target environment using PPO under a fixed transfer horizon. For the teacher trained with energy loss, the $m_{in}$ and $m_{out}$ are set to 12 and 14 respectively. The training is conducted using consistent hyperparameters, as detailed in the next section. All experiments are repeated with 3 random seeds to ensure stability and reproducibility. The transfer horizon varies depending on the setup and source-target configuration. The table below summarizes the number of environment steps used during student training for each case:

Table 3: Transfer horizons (in millions of environment steps) used for student training in each Overcooked setup and configuration. Each experiment is run with 3 random seeds.

| Setup | Recipe ($2 \rightarrow 3$) | Recipe ($1 \rightarrow 3$) | Recipe + Layout ($2 \rightarrow 2$) |
|---|---|---|---|
| Simple Room | 20M | 20M | 12M |
| Ring Room | 35M | 35M | 20M |

## C  Hyperparameters

### C.1  GridWorld

All experimental setups in GridWorld are trained using a fixed set of PPO hyperparameters, summarized in Table 4. These settings remain consistent across all teacher and student training runs within the domain.

### C.2  Overcooked-AI

All Overcooked experiments use a shared set of core PPO hyperparameters, listed in Table 5. These settings are consistent across teacher and student training. However, the learning rate and reward shaping horizon vary depending on the layout and recipe configuration, summarized in Table 6. We use the following notation: O = Onion, T = Tomato, F = Fish, OT = Onion + Tomato, TF = Tomato + Fish, OTF = Onion + Tomato + Fish.

## D  Model Architecture

All MiniGrid experiments share the same model architecture shown in Fig. 10a. Similarly, all Overcooked experiments use the architecture in Fig. 10b. Due to layout size differences in Overcooked, the dense layer input size is set to 182 for *Simple* layouts and 257 for *Ring* layouts.

Table 4: Hyperparameters used for all GridWorld experiments.

| Hyperparameter | Value |
|---|---|
| Learning rate | 0.0005 |
| Discount factor ($\gamma$) | 0.9 |
| GAE lambda ($\lambda$) | 0.8 |
| Policy clip parameter | 0.2 |
| Value function clip parameter | 10.0 |
| Value loss coefficient | 0.5 |
| Entropy coefficient | 0.01 |
| Train batch size | 256 |
| SGD minibatch size | 128 |
| Number of SGD iterations | 4 |
| Number of parallel environments | 8 |
| Normalize advantage | False |

Table 5: Shared PPO hyperparameters across all Overcooked experiments.

| Hyperparameter | Value |
|---|---|
| Discount factor ($\gamma$) | 0.99 |
| GAE lambda ($\lambda$) | 0.6 |
| KL coeff | 0.0 |
| Reward clipping | False |
| Clip parameter | 0.2 |
| VF clip parameter | 10.0 |
| VF loss coeff | 0.5 |
| Entropy coeff | 0.1 |
| Train batch size | 9600 |
| SGD minibatch size | 1600 |
| SGD iterations | 8 |
| Parallel envs | 24 |
| Normalize advantage | False |

Table 6: Setup-specific learning rates and reward shaping horizons.

| Layout | Config | LR | Horizon |
|---|---|---|---|
| Simple | Recipe (O) | 0.001 | 8M |
| | Recipe (OT) | 0.001 | 15M |
| | Recipe (OTF) | 0.001 | 25M |
| | Recipe + Layout (OT) | 0.001 | 10M |
| | Recipe + Layout(TF) | 0.001 | 10M |
| Ring | Recipe (O) | 0.0006 | 10M |
| | Recipe (OT) | 0.0006 | 20M |
| | Recipe (OTF) | 0.0006 | 30M |
| | Recipe + Layout (OT) | 0.0006 | 15M |
| | Recipe + Layout (TF) | 0.0006 | 15M |

## E   ENERGY-BASED LOSS

### E.1   EFFECT OF MARGIN HYPERPARAMETERS ON SEPARATION

We evaluate whether varying the energy thresholds $m_{\text{in}}$ and $m_{\text{out}}$ affects the teacher's ability to distinguish between false and true out-of-distribution (OOD) states. The energy loss used during training is defined over the energy score $\phi(s) = -E(s)$ as:

$$\mathcal{L}_{\text{energy}} = \mathbb{E}_{\mathbf{s}_{\text{in}} \sim \mathcal{D}_{\text{in}}^{\text{train}}} \left[ \left( \max\left(0, m_{\text{in}} - \phi(\mathbf{s}_{\text{in}})\right)\right)^2 \right] + \mathbb{E}_{\mathbf{s}_{\text{out}} \sim \mathcal{D}_{\text{out}}^{\text{train}}} \left[ \left( \max\left(0, \phi(\mathbf{s}_{\text{out}}) - m_{\text{out}}\right)\right)^2 \right].$$

**Experimental Setup**   Experiments are conducted in the *GridWorld (unlocked-to-locked)* environment. During training, the in-distribution (ID) set consists of the most recent 3,000 frames collected from the agent's own trajectory. The out-of-distribution (OOD) set is fixed and sampled from 100 episodes of a random policy in the target environment, where the agent is randomly initialized in any room at the start of each episode to ensure unbiased state coverage (rather than being constrained to the upper room). We evaluate six combinations of $(m_{\text{in}}, m_{\text{out}})$ used in the energy regularization loss (defined over energy scores $\phi(s) = -E(s)$): (10, 15), (5, 10), (15, 20), (10, 10), (15, 15), (12, 14). Each configuration is trained with 3 random seeds using a shared PPO setup and evaluated at the 800,000-step checkpoint.

**Sensitivity Evaluation Protocol.**   We assess whether the teacher consistently distinguishes between *false OOD* states – those similar to ID states and where guidance should be issued – and *true*

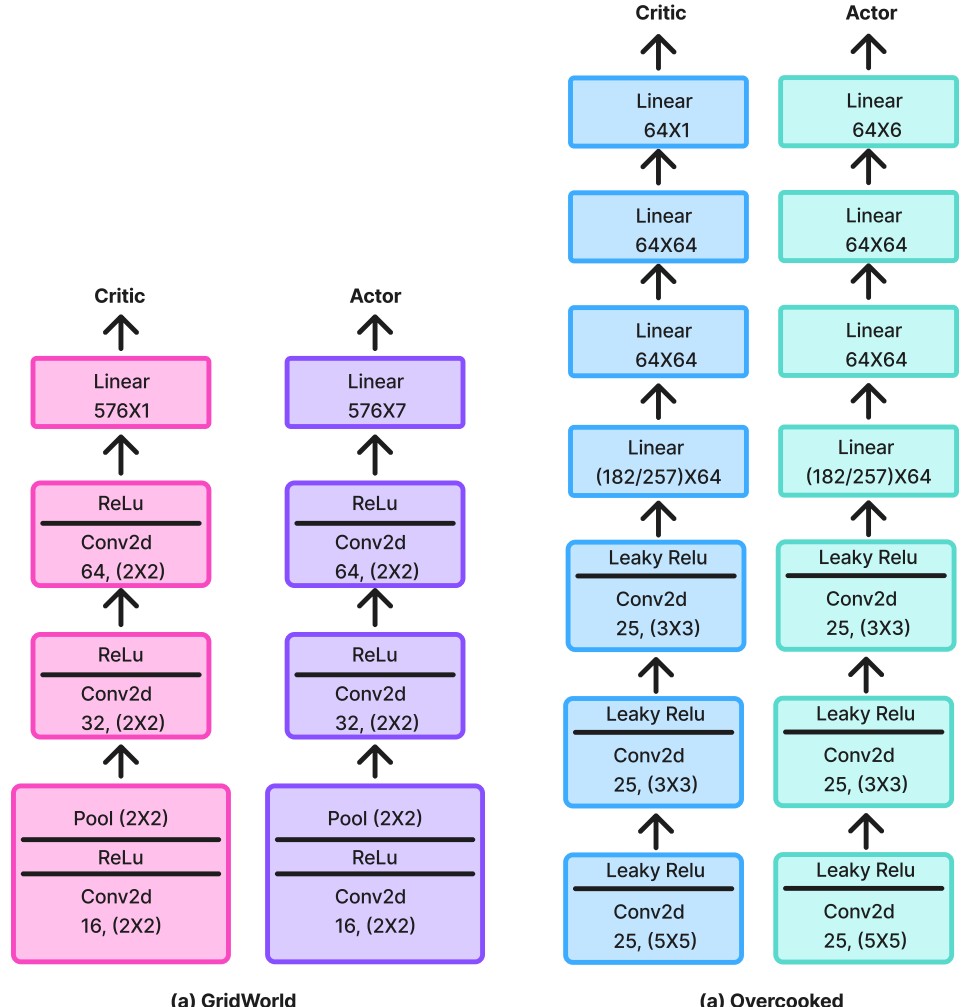

Figure 10: Actor-Critic architectures used in our experiments. (a) MiniGrid. (b) Overcooked.

*OOD* states – those clearly out-of-distribution and where guidance should be withheld. Both sets are drawn from a fixed OOD dataset collected via a random policy in the target environment. For each $(m_{in}, m_{out})$ configuration, we compute the divergence between the energy score distributions of false and true OOD states across three training seeds using Jensen-Shannon divergence, total variation distance, Hellinger distance, and Kullback-Leibler (KL) divergence. To evaluate sensitivity, we apply one-way ANOVA and Kruskal-Wallis tests to determine whether this separation remains consistent across different regularization settings. A high p-value indicates that the teacher's ability to determine when to issue guidance is robust to the choice of $(m_{in}, m_{out})$.

| Metric | ANOVA p-value | Kruskal–Wallis p-value |
|---|---|---|
| Jensen–Shannon | 0.1138 | 0.1592 |
| Kullback–Leibler | 0.2457 | 0.1799 |
| Total Variation | 0.1728 | 0.2322 |
| Hellinger Distance | 0.1247 | 0.1592 |

Table 7: Statistical test results (p-values) for divergence between False OOD and True OOD energy distributions across different $(m_{in}, m_{out})$ settings.

Table 8: Choice of ID states for setting the quantile $q$. "Post-train" derives $q$ from evaluation rollouts of the converged teacher (no exploration); "In-train" derives $q$ from exploration-time states during teacher training. Values are student transfer performance relative to training from scratch (mean $\pm$ 95% CI

.

(a) Alternating-Goal Environment

| Threshold | 0 | 0.1 | 0.2 | 0.3 | 0.4 | 0.5 | 0.6 | 0.7 | 0.8 | 0.9 |
|---|---|---|---|---|---|---|---|---|---|---|
| **Post-train** | $-9.8\pm6.0$ | $21.2\pm8.8$ | $22.5\pm9.0$ | $28.9\pm10.6$ | $33.1\pm8.0$ | $37.8\pm6.5$ | $46.9\pm4.2$ | $37.8\pm8.7$ | $25.9\pm12.0$ | $34.7\pm7.1$ |
| **In-train** | $1.9\pm8.0$ | $36.4\pm6.9$ | $41.6\pm4.3$ | $41.2\pm4.9$ | $46.1\pm8.2$ | $46.1\pm4.5$ | $37.3\pm4.2$ | $40.3\pm4.7$ | $25.8\pm7.4$ | $29.2\pm5.9$ |

(b) UnlockedToLocked Environment

| Threshold | 0 | 0.1 | 0.2 | 0.3 | 0.4 | 0.5 | 0.6 | 0.7 | 0.8 | 0.9 |
|---|---|---|---|---|---|---|---|---|---|---|
| **Post-train** | $11.2\pm4.6$ | $21.5\pm4.5$ | $31.1\pm1.6$ | $29.2\pm4.8$ | $31.7\pm5.2$ | $36.8\pm4.6$ | $40.0\pm2.8$ | $38.0\pm3.3$ | $35.8\pm5.3$ | $24.3\pm3.9$ |
| **In-train** | $16.8\pm4.4$ | $27.1\pm3.8$ | $29.3\pm3.7$ | $31.2\pm4.8$ | $28.8\pm3.1$ | $33.6\pm3.4$ | $35.3\pm2.5$ | $35.1\pm2.7$ | $30.7\pm7.6$ | $32.9\pm5.8$ |

**Results.** As shown in Table 7, we observe no statistically significant variation in the separation between false and true OOD states across different $(m_{in}, m_{out})$ configurations. The ANOVA and Kruskal-Wallis tests yield p-values above 0.1 for all four divergence metrics, indicating that the teacher's ability to distinguish between states where guidance should or should not be issued is stable across regularization settings.

## E.2 CHOICE OF ID STATES

When available, we set the threshold quantile $q$ from the empirical distribution of *in-train* states $\mathcal{D}_{in}^{train}$ collected during teacher on-policy learning (with exploration). When the teacher's training distribution is unavailable, we approximate this *post-train* by rolling out the converged teacher (no exploration) and computing $q$ from those states. Because exploration noise is negligible at convergence, these rollouts serve as a reliable proxy for the high-density regions of the teacher's visitation distribution. We validated both choices by running 100 on-policy evaluation episodes and setting $q$ from the resulting state samples across 10 seeds in two GridWorld settings. Tables 8a–8b report student transfer performance (relative to training from scratch) across quantiles.

## E.3 EFFECT OF ENERGY REGULARIZATION ON TEACHER CONVERGENCE

We assess whether adding the energy regularizer to the *teacher* objective harms final performance or slows learning. Across two GridWorld source tasks and 10 seeds, final returns are unchanged, while convergence is faster with the energy term.

Table 9: Training steps to convergence (mean over 10 seeds; lower is better).

| Teacher | Without energy loss | With energy loss |
|---|---|---|
| Alternating Room | 100,000 | 60,000 |
| Unlocked-to-Locked | 260,000 | 220,000 |

We conjecture that the acceleration arises because the energy term adds an inductive bias that highlights which states are in-distribution (high score) versus out-of-distribution (low score), guiding updates toward familiar regions of the state space more efficiently.

## F DECAY SCHEDULES

This section provides the full comparison between linear decay and several budget–based decay mechanisms. Although both approaches aim to reduce teacher influence over training, we found that linear decay is considerably easier to control and more stable across tasks.

All experiments use the *Unlocked-to-Locked* setting (1M steps). We evaluate five variants:

1. **No Decay**: the teacher issues advice whenever the state is classified as in-distribution.

2. **Single Budget (No Reset)**: a fixed budget equal to 10% of the total steps; once depleted, no further advice is allowed.

3. **Interval Budget (With Reset)**: every 10,240 steps, the teacher is allowed to spend 10% of that interval as advice.

4. **Interval Budget + Linear Decay**: variant (3) with a linear decay factor applied within each interval.

5. **Linear Decay (Ours)**: a single linear decay schedule over training, ending at the midpoint.

Figure 11 reports the performance of all five schedules. The results show that budget-based schemes, with or without reset, often lead to negative transfer when $q < 0.7$. The issue is not budget exhaustion but that a fixed budget allows too much advice early in training. This limits the student's opportunity to act on its own during the exploration phase, which is important for learning the target task. Because the number of early interventions depends on episode structure and exploration behavior, budget schedules are also difficult to tune.

Linear decay avoids this problem by reducing advice gradually and predictably over time. It prevents excessive early intervention while giving the student increasing autonomy as training progresses, without requiring a fixed allowance of advice.

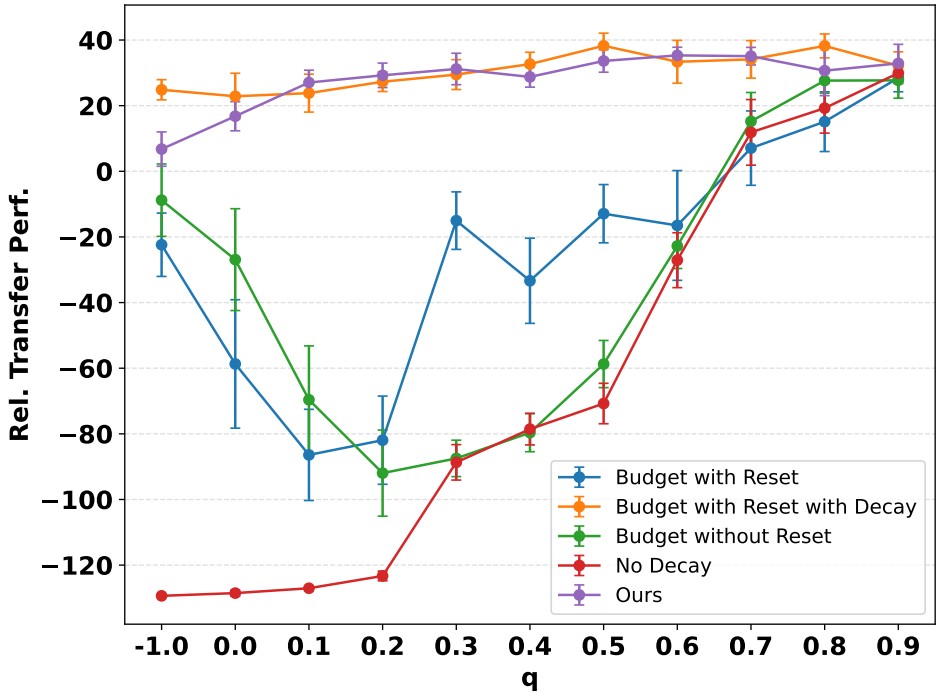

Figure 11: 10 seeds. Comparison of five decay schedules. $q = -1$ advise in all states (AA Baseline).

## G  CHOICE OF OOD BATCHES

The role of $L_{energy}$ is to separate states that lie within the teacher's training distribution from those that do not. In the main paper, OOD states are taken from random rollouts of the student envrionment. This choice was made because they provide a concrete and intuitive illustration of this boundary. To address the concern about future leakage, we conduct a new experiment in the *Unlocked-to-Locked* setting where OOD states are sampled uniformly from the full MiniGrid observation space. This sampling does not assume anything about the target task; it draws from all valid observations that could occur in the domain, and we match the sample size to the original setup for a fair comparison. This removes any possibility of information leakage. Figure 12 reports the results. Performance remains consistent with the main paper, and we continue to observe clear transfer improvements.

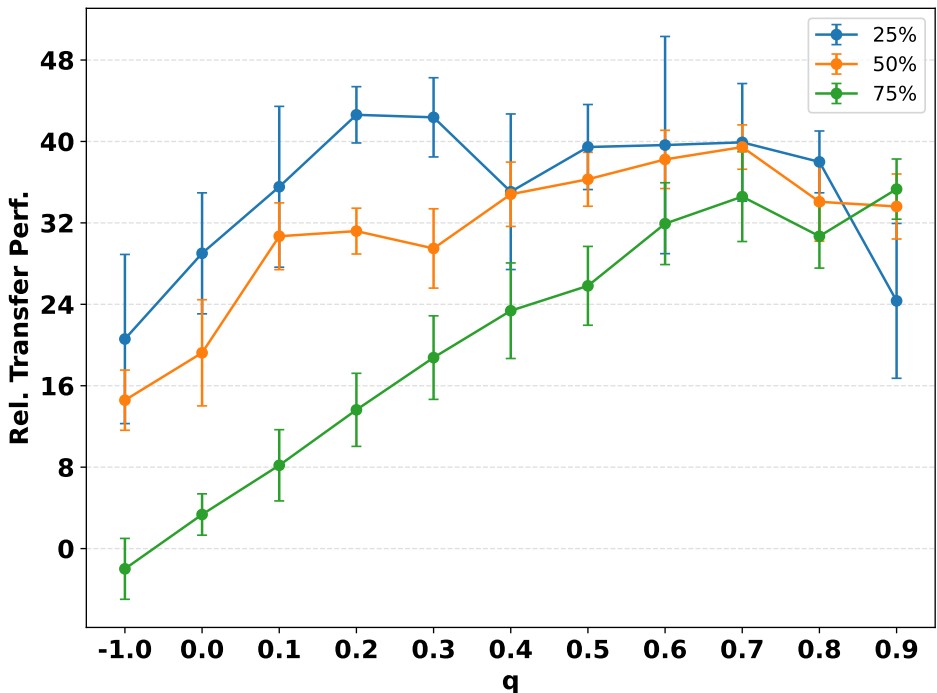

Figure 12: 10 seeds. Effect of replacing rollout-based OOD batches with randomly sampled OOD batches. $q = -1$ advise in all states (AA Baseline).

## H    ADVICE RATES DURING TRAINING

This section reports the rate at which the teacher issues advice and the rate at which the student follows that advice (after applying the decay schedule) in the *Unlocked-to-Locked* environment. We evaluate several values of $q$ under multiple decay schedules (25%, 50%, 75% of the training horizon). The result is shown in  13.

As training progresses, the issue-advice rate tends to increase. Once the student begins to solve more of the task on its own, it encounters states that fall within the teacher's training distribution more frequently, and the teacher identifies a larger fraction of states as familiar. The take-advice rate is the product of the issue-advice rate and the decay factor, so it decreases over time even as the teacher becomes more confident. These measurements describe how the influence of the teacher evolves throughout training and show that EBTL creates a controlled shift from teacher-guided actions to fully student-driven behavior.

## I    ENERGY FORMULATION ON CONTINUOUS ENERGY SCORE

### I.1    CONTINUOUS CONTROL

In robotic control, the action space $\mathcal{A} \subseteq \mathbb{R}^n$ is continuous and $n$-dimensional, with each component $a_i$ corresponding to a separate control command. A common parameterization is a diagonal Gaussian policy, wherein the actor network outputs two vectors $\mu(s) = [\mu_1(s), \ldots, \mu_n(s)]^\top$ and $\sigma(s) = [\sigma_1(s), \ldots, \sigma_n(s)]^\top$, and the policy density is given by

$$p(a \mid s) = \frac{\exp\left(-\frac{1}{2}\left(a - \mu(s)\right)^\top \Sigma(s)^{-1}(a - \mu(s))\right)}{(2\pi)^{n/2} |\Sigma(s)|^{1/2}} \tag{3}$$

where $\Sigma(s) = \mathrm{diag}(\sigma_1^2(s), \ldots, \sigma_n^2(s))$. Equivalently, $\pi_\theta(a \mid s) = \mathcal{N}\big(a; \mu(s), \Sigma(s)\big)$, so each action dimension is sampled as $a_i \sim \mathcal{N}(\mu_i(s), \sigma_i^2(s))$.

An energy-based model defines a joint energy function $E(s, y) \colon \mathcal{S} \times \mathcal{Y} \to \mathbb{R}$, which assigns a scalar energy to each state–output pair $(s, y)$(LeCun et al., 2006). From this one obtains the conditional

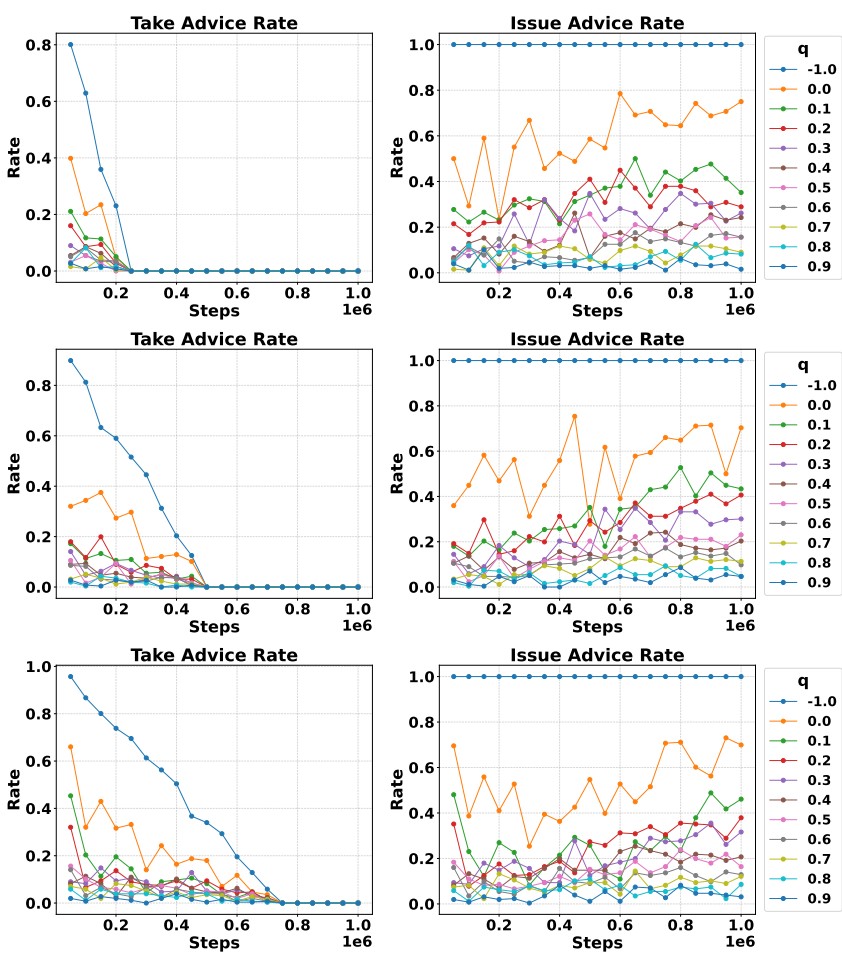

Figure 13: 10 seeds. Advice schedules where advice ends at (a) 25%, (b) 50%, and (c) 75% of training in the Unlocked-to-Locked environment. $q = -1$ advise in all states (AA Baseline).

Gibbs distribution

$$p(y \mid s) = \frac{\exp\big(-E(s, y)\big)}{Z(s)}, \quad Z(s) = \int_{\mathcal{Y}} \exp\big(-E(s, y')\big) \, dy'. \tag{4}$$

where $Z(s)$ is the state-conditional partition function. Here, we derive a continuous-action energy formulation for diagonal Gaussian policies from Equation 3 and Equation 4. We define the joint energy as

$$E(s, a) = \tfrac{1}{2} \, (a - \mu(s))^\top \, \Sigma(s)^{-1} \, (a - \mu(s)). \tag{5}$$

The corresponding state-conditional partition function is

$$Z(s) = \int_{\mathbb{R}^D} \exp\big(-E(s, a')\big) \, da' = (2\pi)^{D/2} \, |\Sigma(s)|^{1/2}. \tag{6}$$

Connecting Equation 5, Equation 6, and Equation 4 and marginalizing out $a$ yields the Helmholtz free energy $E(s) = -\log Z(s) = -\frac{D}{2} \log(2\pi) - \frac{1}{2} \log|\Sigma(s)|$. Since the additive constant $-\frac{D}{2} \log(2\pi)$ does not affect ranking, we drop it and define the simplified energy score as the negative of free energy:

$$\phi(s) = -E(s) = \tfrac{1}{2} \log|\Sigma(s)| = \sum_{i=1}^{D} \log \sigma_i(s). \tag{7}$$

## I.2 ENERGY SCORE AS OOD DETECTOR

**Proposition .2.** *Fix a state $s$, parameters $\theta$, and a* diagonal *Gaussian policy* $\pi_\theta(a \mid s) = \mathcal{N}\big(a; \mu_\theta(s), \Sigma_\theta(s)\big)$ *with* $\Sigma_\theta(s) \succ 0$. *Let $\theta^+$ denote the parameters after one* on-policy, fixed-batch *gradient step (using a batch that includes $s$) on the surrogate below. If visiting $s$ reveals* many com- *patible actions in the sense that $v_{w,i} \geq \sigma_i^2$ for all $i$ and $v_{w,j} > \sigma_j^2$ for some $j$, then $\phi_{\theta^+}(s) > \phi_\theta(s)$.*

*Proof.* Consider the fixed-batch surrogate $\mathcal{L}_s(\theta) = \mathbb{E}_{a \sim \mathcal{D}_s}[w(s, a) \log \pi_\theta(a \mid s)]$, where $\mathcal{D}_s$ is held fixed (e.g., the on-policy batch at $s$) and $w \geq 0$ with $W = \mathbb{E}[w] > 0$. Let $\bar{a}_{w,i} = \mathbb{E}[w \, a_i]/W$ and $v_{w,i} = \mathbb{E}[w \, (a_i - \bar{a}_{w,i})^2]/W$ be the advantage-weighted mean and variance along coordinate $i$.

For a diagonal Gaussian, the mean gradient points from $\mu_i$ toward the weighted sample mean $\bar{a}_{w,i}$; the (log-variance) gradient is positive exactly when the (advantage-weighted) average squared deviation around the *current* mean exceeds the current variance:

$$\frac{\partial \mathcal{L}_s}{\partial \mu_i} = \frac{W}{\sigma_i^2} (\bar{a}_{w,i} - \mu_i), \qquad \frac{\partial \mathcal{L}_s}{\partial \theta_i} = \frac{W}{2} \left( \underbrace{\frac{v_{w,i} + (\bar{a}_{w,i} - \mu_i)^2}{\sigma_i^2} - 1}_{\Delta_i} \right), \quad \text{where } \theta_i = \log \sigma_i^2.$$

After a step with stepsize $\eta > 0$, the mean moves toward the (advantage-)weighted mean, and each variance toward the (advantage-)weighted spread about the current mean:

$$\mu_i^+ = \mu_i + \eta \frac{W}{\sigma_i^2} (\bar{a}_{w,i} - \mu_i), \qquad \theta_i^+ = \theta_i + \eta \frac{W}{2} \Delta_i \implies \sigma_i^{2+} = \sigma_i^2 \exp\big[\eta \frac{W}{2} \Delta_i\big].$$

Thus, $\sigma_i^{2+} - \sigma_i^2$ has the same sign as $\Delta_i$: the update increases (resp. decreases) $\sigma_i^2$ whenever the observed average squared deviation at $s$ is larger (resp. smaller) than the model's current variance. By the proposition's premise ("many compatible actions"), the observed advantage-weighted dispersion is nontrivial: $v_{w,i} \geq \sigma_i^2$ for all $i$ and $v_{w,j} > \sigma_j^2$ for at least one coordinate $j$; since $(\bar{a}_{w,i} - \mu_i)^2 \geq 0$, this implies the above increase condition. Hence $\sigma_i^{2+} \geq \sigma_i^2$ for all $i$ with strict increase for some $j$, so $\sum_i \log \sigma_i^{2+} > \sum_i \log \sigma_i^2$. As $\phi_\theta(s) = \sum_i \log \sigma_i^2$, it follows that $\phi_{\theta^+}(s) > \phi_\theta(s)$. $\square$

**Frequently visited states receive higher scores.** We initialize the neural network parameters such that the initial variance for each state is close to 1. Given that the parameters are randomly initialized, initial estimates of $\mu(s)$ are typically inaccurate. Since the variance update is driven by the square residual $(a - \mu(s))^2$, early in training the residuals are large and each visit to state $s$ drives the

variance up. Thus, states that are visited more frequently yield higher variance estimates which result in higher energy scores. Furthermore, in continuous action spaces many tasks exhibit some degree of aleatoric uncertainty, i.e. many actions are "equally good". This results in a non-zero variance even after policy convergence, the true value of which depends on the spread of compatible actions at state $s$.

