# OpenReview forum: "Energy-Based Transfer for Reinforcement Learning"
_ICLR.cc/2026/Conference — Submitted to ICLR 2026_

### Official Review · Reviewer_Y6hM · 2025-10-24

**Soundness:** 2
**Presentation:** 2
**Contribution:** 3
**Rating:** 4
**Confidence:** 3

**Summary:**

This paper addresses the low sample efficiency of reinforcement learning and the "suboptimal guidance" flaw of traditional transfer learning. It proposes Energy-Based Transfer Learning (EBTL): using energy scores to detect in-distribution or out-of-distribution states. The teacher only guides the student in ID states, while OOD states let the student explore freely. Energy regularization is added to enhance ID/OOD separability without harming teacher performance. Experiments on single-task and multi-task scenarios show EBTL outperforms 5 baselines in sample efficiency and final rewards. Theoretically, energy scores are proven proportional to the teacher’s state visitation density, ensuring reliable guidance decisions.

**Strengths:**

1. The core insight is good. The paper ties the timing of intervention to the teacher’s actual competence, ensuring guidance is only deployed when it adds value.
2. The experiments target the exact pain points EBTL aims to solve. The team tests on both Minigrid navigation and Overcooked cooking scenarios, two settings where knowledge transfer can facilitate exploration. And the results show that EBTL outperforms all baselines.

**Weaknesses:**

1. A notable limitation of EBTL lies in its narrow reliance on the current single-state representation to calculate energy scores. It overlooks the critical role of the agent’s impending actions and historical state sequences when evaluating whether knowledge transfer is appropriate. This oversimplification may lead to misjudgments about "guidance value" in scenarios where state meaning depends on context.
2. The paper also fails to explore obvious, promising extensions that would use the energy score’s graduality more effectively. It ignores the energy score’s inherent value as a continuous measure of familiarity and skips more adaptive approaches like mixed action weighting or importance-guided selection. Instead of a binary “guide or not,” use the energy score to directly weight the mix of teacher and student actions. A higher score could mean a higher probability of sampling the teacher’s action. Besides, since the energy score correlates with the teacher’s state visitation density, it could refine the importance weights EBTL already uses to correct offline bias.
3. I want the authors to consider more practical transfer settings like sim2real.

**Questions:**

See weaknesses.

---

> ### Author Response · Authors · 2025-11-21
> **Response**
>
> Thank you for your thoughtful review. We have uploaded a revised version of the paper, with all newly added sections marked in red. Section 5.3 includes new experiments in a continuous action space. Appendix F compares budget-based decay with linear decay. Appendix G presents an alternative choice of OOD batches. Appendix H reports how the advice rate changes during training. Appendix I provides the formulation of the continuous energy score.
>
> ### weakness1
> **Our method does not overlook temporal context.** Under the MDP formulation, any task history that is relevant for decision-making can be encoded directly into the state, and EBTL simply evaluates the representation that the teacher policy already uses. Prior work such as [1] encodes a state using multiple time steps, and EBTL is fully compatible with this design. Conditioning on past trajectories does not break the Markov property as long as the history information is included as part of the state vector.
>
> In our transfer setting, the teacher is assumed to have converged on its own task and gives advice only on states it knows how to handle. When advice is issued on states the teacher has seen during training, the student reaches rewarding states more often than if relying solely on random exploration. This is why selective transfer is appropriate here.
>
> **When task context changes, this information can also be included in the state.** Our Overcooked setup illustrates this: although two scenes may appear the same, the observation embedding includes the current recipe the agent is working on (onion, tomato, or fish), so the contextual meaning of the state is explicitly represented. EBTL evaluates energy on this enriched state representation, so it naturally incorporates context rather than ignoring it.
>
>
> ### weakness2
> Our setting operates directly on environment actions, so teacher–student interaction is inherently discrete. At each step, the agent must either take the teacher’s action or take its own; there is no meaningful way to “mix” two actions. For example, combining “go left” and “go right” would not create a valid or useful command in grid-world navigation and would often lead to collisions or infeasible behavior. Since actions are executed in the environment rather than in a latent space, partial interpolation simply does not apply.
> EBTL therefore asks a binary question: does the teacher know this state well enough to advise? If yes, we use the teacher’s action; if not, the student proceeds on its own. This matches the structure of action-level control in the environments we study, where a state is either familiar to the teacher or not. For this reason, soft action-mixing or gradual weighting does not fit our setting. Our binary gating reflects the actual semantics of executing actions in these tasks.
>
>
> ### weakness3
> Our goal in this paper is to isolate and demonstrate the effect of selective transfer based on OOD awareness. The settings we study already show that filtering advice by distribution improves sample efficiency and avoids negative transfer, which is the central claim we aim to establish.
>
> In the revised version, we also added Section 5.3 with a continuous-action environment and continuous observations. This setup is closer to practical scenarios and shows that the mechanism is not restricted to discrete grids. While sim-to-real involves additional engineering challenges beyond the scope of this paper, the new experiments strengthen the evidence that selective transfer based on OOD detection remains effective once we move beyond discrete domains.
>
> [1] Chen, Lili, et al. "Decision transformer: Reinforcement learning via sequence modeling." Advances in neural information processing systems 34 (2021): 15084-15097.

---

### Official Review · Reviewer_7nhW · 2025-10-26

**Soundness:** 3
**Presentation:** 3
**Contribution:** 3
**Rating:** 6
**Confidence:** 2

**Summary:**

This paper introduces energy-based transfer learning (EBTL), which improves sample efficiency in reinforcement learning through selective teacher guidance. EBTL employs the teacher’s energy as a familiarity proxy, issuing advice only in likely in-distribution states and thereby avoiding additional networks, mappings, or handcrafted OOD detectors. Empirically, EBTL outperforms baselines in both single- and multi-task transfer.

**Strengths:**

1. Originality:
The paper introduces a novel energy-based mechanism to control knowledge transfer in RL. While energy models have been used in OOD detection, applying them to decide when to transfer across tasks is an original and creative idea.

2. Quality:
The technical development is sound and theoretically grounded. The algorithm design is coherent with the underlying theory. Experiments are comprehensive, covering both single- and multi-task settings. The results consistently demonstrate the method’s efficiency and robustness.

3. Clarity:
The paper is clearly written and well-organized. The motivation, methodology, and theoretical insights are presented logically, with helpful figures and algorithm descriptions.

4. Significance:
EBTL addresses a core limitation in transfer RL—negative transfer under distributional shift—and provides a general framework applicable to various domains. Its ability to adaptively decide when to transfer makes it both practically relevant and conceptually impactful, offering a promising direction for improving sample efficiency in multi-task and continual RL.

**Weaknesses:**

1. Limited theoretical depth:
The link between energy scores and teacher visitation density is only intuitively discussed, without formal guarantees on convergence or optimality. Providing stronger theoretical analysis—e.g., on transfer efficiency or sample complexity—would enhance rigor.

2.Incomplete ablation analysis:
The effects of key components (energy threshold τ, decay schedule, regularization) are not fully disentangled. More systematic ablations would clarify each component’s contribution and the model’s behavior over training.

**Questions:**

1. Scalability to complex domains:
 Could the authors discuss whether EBTL can handle continuous-action tasks ? Are there computational issues when estimating energy scores in such settings?

2. Robustness to poor teachers:
While the paper claims robustness to imperfect teachers, there is little quantitative evidence. How does EBTL perform when the teacher policy is partially suboptimal or even misleading? Could the authors include or elaborate on experiments that vary teacher quality?

---

> ### Author Response · Authors · 2025-11-21
> **Response**
>
> Thank you for your thoughtful review. We have uploaded a revised version of the paper, with all newly added sections marked in red. Section 5.3 includes new experiments in a continuous action space. Appendix F compares budget-based decay with linear decay. Appendix H reports how the advice rate changes during training. Appendix I provides the formulation of the continuous energy score.
>
> ### weakness1
> We do give a formal link between energy and visitation. Proposition 4.1 states the exact condition under which the policy’s logits define a valid free-energy model, and Appendix A.1 proves that a standard on-policy update increases the score on states that are visited more often. This means the connection is not only intuitive: the update rule itself pushes energy to track visitation. For transfer, the key quantity is how often the teacher gives correct guidance so that the student reaches rewarding states more reliably. Figures 3 and 4 measure this directly and show that correct guidance dominates. Taken together, Proposition 4.1 and Appendix A.1 already give the theoretical justification needed for why energy reflects teacher familiarity in the way our method uses it.
>
> ### weakness2
> **Ablation on Energy Threshold.** Removing energy gating corresponds to always accepting teacher actions. This case is exactly the Action Advising baseline we include, and it is also represented as the setting \\( q = -1 \\) in Figure 3(a) and Figure 4(a). The performance of this no-gating variant is shown in Figure 3, Figure 4, and Figure 6, where Action Advising consistently underperforms EBTL, indicating that decay alone is not sufficient. These baselines use the same linear decay schedule as our method, which allows a clean comparison and shows the impact of energy gating.
>
> **Ablation on Decay Schedule.** A decay mechanism is necessary in online transfer learning because it allows the student to act on its own in the later phases of training rather than relying indefinitely on the teacher. The baselines we include, such as Jumpstart RL and Kickstart RL, also depend on this decay for exactly this reason. To illustrate its importance, we did a *new ablation in Appendix.F* that removes the decay entirely, so the student accepts teacher actions whenever the teacher issues advice. As shown in the results, this variant leads to negative transfer when \\( q < 0.7 \\).
>
> **Ablation on Regularization.** Figure 5(a) and lines 358–362 already evaluate the effect of the energy regularizer. The energy loss consistently improves EBTL, especially in the Locked-Room setting where the covariate shift is larger. At the same time, even without the energy loss, EBTL still outperforms all baselines. This follows from the fact that our decision rule is tied to the teacher’s state-visitation distribution, so the method is already well-grounded; the regularizer simply strengthens this separation under larger shifts.
>
>
>
> ### question1
> EBTL is not limited to discrete-action tasks. If a continuous action space is discretized, as done in systems like OpenVLA[1], our discrete energy-score formulation applies directly without any modification. Several reviewers asked about continuous control, so we added a new experiment in Section 5.3 of the revised paper. This setup uses a continuous-state, continuous-action environment and shows that the transfer mechanism carries over cleanly. The energy score is straightforward to compute in this setting, and we did not encounter computational issues when estimating it.
>
>
> ### question2
> There are two forms of suboptimality to consider. Throughout the paper we assume the teacher is optimal in its own task, but we also include experiments with weaker teachers because they highlight the role of selective transfer. Table 1 and Section 5.1 train teachers at different proficiency levels and show that EBTL consistently improves as teacher quality increases; even weaker teachers still offer occasional useful guidance that accelerates progress, and the student only follows the teacher in a subset of states, so the effect of imperfect advice is limited and transfer remains positive. This monotonic trend does not appear in baselines that imitate indiscriminately, which often fail to benefit from stronger teachers or even deteriorate due to task misalignment, reinforcing the need for energy-based gating.
>
> The second form of suboptimality is approximation error in separating in-distribution from out-of-distribution states. Figures 3 and 4 report the fraction of correct and incorrect teacher suggestions across q values. Although the teacher occasionally misclassifies OOD states, it still issues far more correct than misleading suggestions, and q controls this trade-off. As long as correct guidance dominates, EBTL remains robust, which is what we observe across all experiments.
>
> [1]. Kim, Moo Jin, et al. "Openvla: An open-source vision-language-action model." arXiv preprint arXiv:2406.09246 (2024).

---

> > ### Comment · Reviewer_7nhW · 2025-11-25
> >
> > Thanks for your detailed rebuttal. I have carefully reviewed your comments and will keep my initial score.

---

> > > ### Author Response · Authors · 2025-11-25
> > >
> > > Thank you for taking the time to read our rebuttal. We’ve tried to follow up on each of the concerns raised, and it would be helpful if you could clarify which parts you still feel are unresolved. Any specifics you can share would help us understand your perspective and improve the work.

---

### Official Review · Reviewer_uXHE · 2025-10-31

**Soundness:** 2
**Presentation:** 4
**Contribution:** 2
**Rating:** 4
**Confidence:** 3

**Summary:**

This paper addresses the challenge of low sample efficiency in reinforcement learning (RL), particularly in multi-task and continual learning settings. The authors propose an energy-based transfer learning framework that leverages a previously trained teacher policy to guide exploration in new tasks. To avoid negative transfer when the target task diverges from the teacher’s domain, the method employs out-of-distribution detection based on energy scores, ensuring that teacher interventions occur only in familiar states. The authors provide a theoretical justification linking energy scores to the teacher’s state-visitation density and present empirical results demonstrating that the approach improves both sample efficiency and final performance across single-task and multi-task benchmarks.

**Strengths:**

* This work tackles a good problem, in real world settings, it has been shown that with respect to a reward function the teacher may be considered sub-optimal in parts of the task.
* The authors discuss between same task transfer, and mult-task transfer, which makes allows insight into both perspectives, unlike other works which may focus on one or the other.
* The performance seems to be good on two applicable grid based environments and the authors clearly show that their method has advantages over the other advising algorithms.
* The paper is well written, and the diagrams chosen make things clear to the reader.

**Weaknesses:**

Recently, there have been ways introduced to see if a policy is out of distribution specific to the RL domain, see [1,2,3,4], the energy function may not work in OOD scenarios, potentially in partially observable environments, and it might be necessary to take insight from [1,2,3,4] or discuss the expected changes from a unsupervised ood method to a suitable RL ood method. This may provide insight to section ```Higher covariate shift makes OOD detection more challenging ```.

Although it is an older work and is focused on sub-optimal ensembles of teachers, I was surprised that the authors did not have much insight on the issues [5] encountered, specifically one of the main takeaways:
* Behavioral policies can be sensitive to contradictory teachers
  - When two policies contradict, i.e potentially the teacher and the student, there are works that find it very difficult for the task to be completed [6], intuitively it may find itself stuck in a state as one policy recommends to leave, where the other (potentially student) keeps recommending to return. There does not seem to be any explanation to why the authors would not encounter this issue.

The decay parameter seems to be very important to the task, yet rarely talked about. Unlike other works [7], t is not budget based. One scenario that comes to mind is to suppose the teacher is considered OOD for the first 3 quarters of the task. It appears that as the horizon increases, the teacher will never be called to do any action advising depending on the decay and would reduce to a single agent. There are pros and cons of using a horizon based decay vs a budget based decay, however there doesn't seem to be any talking points on this either, perhaps ablation experiments should be added? Why was decay chosen instead of budget?


[1] Mohamad H Danesh and Alan Fern. Out-of-distribution dynamics detection: Rl-relevant benchmarks and results. arXiv preprint arXiv:2107.04982, 2021.

[2] Hongming Zhang, Ke Sun, Linglong Kong, Martin Muller, et al. A distance-based anomaly detection framework for deep reinforcement learning. Transactions on Machine Learning Research,
2024.

[3] Geigh Zollicoffer, Kenneth Eaton, Jonathan C Balloch, Julia Kim, Wei Zhou, Robert Wright, & Mark Riedl (2025). Novelty Detection in Reinforcement Learning with World Models. In Forty-second International Conference on Machine Learning.

[4] Tom Haider, Karsten Roscher, Felippe Schmoeller da Roza, and Stephan Gunnemann. Out-of- ¨
distribution detection for reinforcement learning agents with probabilistic dynamics models. In
Proceedings of the 2023 International Conference on Autonomous Agents and Multiagent Systems, pp. 851–859, 2023.

[5] Andrey Kurenkov, Ajay Mandlekar, Roberto Martin-Martin, Silvio Savarese, & Animesh Garg. (2019). AC-Teach: A Bayesian Actor-Critic Method for Policy Learning with an Ensemble of Suboptimal Teachers.

[6] Shenfeld, I., Hong, Z.W., Tamar, A., & Agrawal, P. (2023). TGRL: An Algorithm for Teacher Guided Reinforcement Learning. In Proceedings of the 40th International Conference on Machine Learning (pp. 31077–31093). PMLR.

[7] Da Silva, F. L., Hernandez-Leal, P., Kartal, B., & Taylor, M. E. (2020). Uncertainty-Aware Action Advising for Deep Reinforcement Learning Agents. Proceedings of the AAAI Conference on Artificial Intelligence, 34(04), 5792-5799.

**Questions:**

Although I don't believe it is necessary due to the increased difficulty of the analysis, but have the authors considered continuous action spaces or continuous state spaces? I would be willing to increase my score if these concerns were adequately addressed.

---

> ### Author Response · Authors · 2025-11-21
> **Response - Part1**
>
> Thank you for your thoughtful review. We have uploaded a revised version of the paper, with all newly added sections marked in red. Section 5.3 includes new experiments in a continuous action space. Appendix F compares budget-based decay with linear decay. Appendix G presents an alternative choice of OOD batches. Appendix H reports how the advice rate changes during training. Appendix I provides the formulation of the continuous energy score.
>
> We agree that OOD detection in RL has been explored from multiple directions, and the cited works [1–4] provide examples of different mechanisms for identifying states outside a policy’s training distribution. EBTL instead focuses on estimating whether a state is familiar to the teacher using only the observation and the teacher’s logits. This is aligned with our transfer setting, where higher covariate shift makes OOD detection harder because observations may deviate substantially from those seen by the teacher. Below, we explain how each cited approach differs in assumptions or objectives from EBTL.
>
> **[1]** considers OOD settings where the raw observation is unchanged while the MDP dynamics shift (the transition function changes). EBTL relies on an assumption that any dynamics change relevant to action selection is ultimately reflected in the observation encoding. We argue that this assumption is not strong in our settings: in Overcooked, for example, when scenes look the same, the observation embedding contains whether the agent is in an onion, tomato, or fish-soup context (the task the agent is handling), so task-relevant MDP differences are captured at the observation level.
>
> **[2]** detects OOD states using distances in internal representation layers. Our use of the energy score is not a generic choice because Proposition 4.1 shows that policy energy is tied to the RL state-visitation distribution, which is the quantity needed for selective transfer. This avoids choosing intermediate layers or storing large sets of embeddings, which can vary by architecture and training setup. Paper [2] requires fitting a class-conditional Gaussian model by computing a mean and covariance matrix for the penultimate-layer embeddings of every action class. The energy score, in contrast, provides a direct and lightweight measure of teacher familiarity without estimating such per-class densities. EBTL only stores a small set of IID energy values, which is stable across architectures and inexpensive to maintain.
>
> **[3]** detects OOD by training a world model and using its prediction errors as the signal. This direction addresses a different objective because it requires learning a separate dynamics model and evaluating discrepancies in predicted transitions. Our work does not involve modeling environment dynamics; the goal is to estimate whether a state lies within the teacher’s visitation distribution. The energy score provides this directly through the teacher policy without introducing an additional world-model component.
>
> **[4]** detects OOD by learning a probabilistic dynamics model and using uncertainty over predicted transitions. Our setting does not involve learning a dynamics model, since EBTL operates directly on the teacher policy. Although Paper [4] targets shifts in transition dynamics, we assume such changes appear in the observation space, which is a standard assumption in model-free RL. Under this assumption, the energy score provides a simple way to assess whether a state is familiar to the teacher, without maintaining a separate dynamics model.
>
>
> The contradictory-policy problem is exactly what selective transfer is designed to handle. Our motivation is that teacher advice can slow or block learning when it conflicts with what the student must do without filtering. This is clear in our baselines: in the Alternating Goal Room task, Action Advising performs poorly because the teacher always suggests going toward Room 1 even though the student must go to Room 3 half of the time; in the Unlocked-to-Locked task, the teacher moves toward the lower room when the student needs to pick up the key, sending the agent in the wrong direction. These are precisely the cases where contradictory advice would trap the agent, and EBTL avoids this by filtering such advice through the energy gate. The student never blends teacher and student actions, and it follows the teacher only when the state is within the teacher’s distribution. In those cases, the advice is aligned with the teacher’s experience and  helps the student.
>
> Due to the limited space, we will address the rest in the next response.

---

> ### Author Response · Authors · 2025-11-21
> **Response - Part2**
>
> We appreciate the reviewer’s insightful question regarding the choice of linear decay versus a budget-based schedule. In fact, our project initially used a budget mechanism, since both approaches share the same goal: ensuring that, as training progresses, the student increasingly acts on its own. We later switched to linear decay because we found it substantially simpler to control and more stable across tasks. To clarify this design choice, we now include **new ablations in Appendix F** using the Unlocked-to-Locked environment (1M steps) that compare five variants:
>
> 1. **No Decay**: the teacher issues advice whenever the state is classified as in-distribution.
> 2. **Single Budget (No Reset)**: a fixed budget equal to 10\% of the total steps; once depleted, no further advice is allowed.
> 3. **Interval Budget (With Reset)**: every 10240 steps, the teacher is allowed to spend 10\% of that interval as advice.
> 4. **Interval Budget + Linear Decay**:  variant (3) with a linear decay factor applied within each interval.
> 5. **Linear Decay (Ours)**:  a single linear decay schedule over training, ending at the midpoint.
>
> You can also see the results here in the table below:
> |name|-1.0|0.0|0.1|0.2|0.3|0.4|0.5|0.6|0.7|0.8|0.9|
> |----|----|----|----|----|----|----|----|----|----|----|----|
> |1|$-129.4\pm0.0$|$-128.5\pm0.9$|$-127.1\pm0.8$|$-123.3\pm1.5$|$-88.7\pm5.4$|$-78.6\pm4.8$|$-70.8\pm6.2$|$-27.1\pm8.4$|$11.9\pm10.0$|$19.2\pm7.6$|$29.9\pm3.2$|
> |2|$-8.8\pm11.0$|$-26.9\pm15.5$|$-69.6\pm16.4$|$-92.0\pm13.1$|$-87.5\pm5.6$|$-79.6\pm5.8$|$-58.7\pm7.2$|$-22.8\pm6.9$|$15.2\pm8.8$|$27.6\pm3.8$|$27.7\pm5.4$|
> |3|$-22.4\pm9.6$|$-58.7\pm19.6$|$-86.4\pm13.9$|$-81.9\pm13.4$|$-15.0\pm8.8$|$-33.4\pm13.0$|$-12.9\pm8.9$|$-16.5\pm16.7$|$7.1\pm11.3$|$15.1\pm9.1$|$28.5\pm4.2$|
> |4|$24.9\pm3.1$|$22.9\pm7.0$|$23.8\pm5.8$|$27.3\pm3.0$|$29.5\pm4.6$|$32.7\pm3.6$|$38.2\pm3.8$|$33.4\pm6.5$|$34.1\pm5.7$|$38.2\pm3.6$|$32.2\pm4.2$|
> |5|$6.8\pm5.2$|$16.8\pm4.4$|$27.0\pm3.8$|$29.3\pm3.7$|$31.2\pm4.8$|$28.8\pm3.1$|$33.6\pm3.4$|$35.3\pm2.5$|$35.1\pm2.7$|$30.7\pm7.6$|$32.9\pm5.8$|
>
> The results show that budget-based schemes, with or without reset, often lead to negative transfer when $q < 0.7$. The issue is not budget exhaustion but that a fixed budget allows too much advice early in training. This limits the student’s opportunity to act on its own during the exploration phase, which is important for learning the target task. Because the number of early interventions depends on episode structure and exploration behavior, budget schedules are also difficult to tune.
>
> Linear decay avoids this problem by reducing advice gradually and predictably over time. It prevents excessive early intervention while giving the student increasing autonomy as training progresses, without requiring a fixed allowance of advice.
>
>
> ### Question1
> For continuous action spaces, one straightforward option is to discretize the action space as in OpenVLA [1], in which case the method becomes identical to our proposed EBTL setup. Alternatively, rather than discretizing, one can define an energy score directly in the continuous setting. We have explored this direction as part of a separate project. Since several reviewers asked about this, we now include the continuous-action version in Section 5.3 of the revised paper, where we evaluate our method in a continuous-action, continuous-state environment. This experiment is closer to practical robotic applications and demonstrates that our approach is not limited to discrete grids.
>
> [1]. Kim, Moo Jin, et al. "Openvla: An open-source vision-language-action model." arXiv preprint arXiv:2406.09246 (2024).

---

### Official Review · Reviewer_5bd8 · 2025-11-01

**Soundness:** 3
**Presentation:** 3
**Contribution:** 2
**Rating:** 4
**Confidence:** 4

**Summary:**

This paper tackles the problem of improving teacher–student transfer in reinforcement learning. The proposed method uses an energy score to decide whether the current state lies within the teacher’s training distribution. If the score exceeds a threshold, the teacher provides action guidance; otherwise, the student acts independently.

**Strengths:**

1. The transfer learning in RL is an important and timely topic.
2. The proposed approach is simple yet effective, and conceptually easy to follow.
3. The paper is clearly written and well structured.

**Weaknesses:**

1. The central theoretical claim (Proposition 4.1) states that the logarithm of the stationary visitation density is proportional to the negative free energy $\phi(s)$. This relies on a very strong assumption, that is, the policy network optimized for reward maximization (e.g., PPO) implicitly forms a *realizable energy model* $p_{\theta^*}$ that perfectly fits the visitation distribution. In practice, the policy’s logits are trained for control, not density estimation, so equating them with a likelihood model is conceptually weak.
2. The student’s training data mixes *on-policy* samples from itself (sampled in training by current policy) and *off-policy* samples from the teacher, whose policy arises from a different distribution. The paper doesn't address how this distribution mismatch is handled or whether importance weighting or correction is applied. This omission raises concerns about biased gradient estimates and unstable policy updates.
3. The experiments are limited to MiniGrid and Overcooked, where the ID–OOD distinction is relatively trivial (goal position or recipe type). These toy setups do not convincingly demonstrate that an energy-based metric is necessary, e.g., simple rule-based distinctions might achieve the same effect. Evaluation in more complex or continuous-control environments would strengthen the paper.

**Questions:**

1. The paper introduces a decay schedule $\delta(t)$ that gradually reduces teacher intervention, similar to $\epsilon$-greedy annealing. However, this design blurs attribution: performance gains may come from the decay itself rather than from energy-based selection. Could the authors run ablation studies removing the decay, or removing the energy gating while keeping only the decay, to isolate the contribution of each component?
2. Regarding the introduction of $L_{energy}$ to augment teacher training, it uses both in-distribution $D_{in}$ and out-of-distribution $D_{out}$ samples. This raises some questions: does this introduce a “future leakage” paradox? That is, since we can sample OOD data from new environments the teacher model has never encountered, does this imply the teacher model has received specialized training in these “unseen” environments beforehand? Does this contradict the principle of transfer learning, which emphasizes learning effectiveness in completely unseen environments?
3. Could the author provide the proportion of teacher intervention during training? I wonder what percentage of actions are sampled from the teacher model, to better understand how crucial the teacher's role truly is. If the proportion at different training stages could be provided, it would allow us to explore how teacher intervention evolves throughout the training process.

---

> ### Author Response · Authors · 2025-11-21
> **Part1 - Response to the Weaknesses**
>
> Thank you for your thoughtful review. We have uploaded a revised version of the paper, with all newly added sections marked in red. Section 5.3 includes new experiments in a continuous action space. Appendix F compares budget-based decay with linear decay. Appendix G presents an alternative choice of OOD batches. Appendix H reports how the advice rate changes during training. Appendix I provides the formulation of the continuous energy score.
>
> ### Weakness1
> Proposition 4.1 is intended as a high-level motivation for why the energy score correlates with state familiarity: it shows that, under an idealized setting, the negative free energy ranks states in the same way as the visitation density. Our method does not assume that PPO learns a perfect density model. For practical purposes, all we require is a weaker property: states visited more frequently by an on-policy agent tend to receive more stable and higher energy score. Appendix A.1 gives a formal argument showing that on-policy updates increase the score on visited states, which supports our use of energy as a practical indicator of in-distribution behavior. Empirically, this approximate relationship is all we need for reliable gating.
>
> ### Weakness2
> As noted in the paper (lines 214 to 215), *“We also incorporate importance-ratio correction, as teacher actions  \\(a_t \sim \pi_T(\cdot \mid s_t)\\) are off-policy for the student.”* In our code, this is handled through the standard importance weight \\(
> w_t = \frac{\pi_S(a_t \mid s_t)}{\pi_S^{\text{old}}(a_t \mid s_t)},\\) which corrects for the distribution mismatch whenever a teacher action enters the student’s buffer.
>
> ### Weakness3
> Although MiniGrid and Overcooked allow clear visualizations of ID and OOD behavior, the distinction in our tasks is not hand-engineered. Rule-based detectors would require access to environment-specific encodings and manual definitions of what counts as in-distribution, while EBTL infers this automatically from the teacher’s policy without any task-specific knowledge. In the Unlocked-to-Locked transfer, the door is always present in the student environment, yet the teacher is still expected to provide reliable advice in post-key states even though neither the door nor the key appear in its training distribution. This setting creates a nontrivial distribution shift that directly tests whether the method can identify when the teacher remains competent. Finally, Section 5.3, which we have newly added, includes a continuous-action experiment showing that the method also extends beyond discrete-action domains.

---

> ### Author Response · Authors · 2025-11-21
> **Response to the Questions**
>
> ### Q1
> Removing energy gating corresponds to always accepting teacher actions. This case is exactly the Action Advising baseline we include, and it is also represented as the setting \\( q = -1 \\) in Figure 3(a) and Figure 4(a). The performance of this no-gating variant is shown in Figure 3, Figure 4, and Figure 6, where Action Advising consistently underperforms EBTL, indicating that decay alone is not sufficient. These baselines use the same linear decay schedule as our method, which allows a clean comparison and shows the impact of energy gating.
>
> A decay mechanism is necessary in online transfer learning because it allows the student to act on its own in the later phases of training rather than relying indefinitely on the teacher. The baselines we include, such as Jumpstart RL and Kickstart RL, also depend on this decay for exactly this reason. To illustrate its importance, we did a *new ablation in Appendix.F* in the *Unlocked-to-Locked* setting that removes the decay entirely, so the student accepts teacher actions whenever the teacher issues advice. As shown in the results, this variant leads to negative transfer when \\( q < 0.7 \\). You can also see it here in the table below ($q=-1$ is the Action Advising Baseline):
>
> | name | -1.0 | 0.0 | 0.1 | 0.2 | 0.3 | 0.4 | 0.5 | 0.6 | 0.7 | 0.8 | 0.9 |
> | ---- | ---- | ---- | ---- | ---- | ---- | ---- | ---- | ---- | ---- | ---- | ---- |
> | No Decay | $-129.4\pm0.0$ | $-128.5\pm0.9$ | $-127.1\pm0.8$ | $-123.3\pm1.5$ | $-88.7\pm5.4$ | $-78.6\pm4.8$ | $-70.8\pm6.2$ | $-27.1\pm8.4$ | $11.9\pm10.0$ | $19.2\pm7.6$ | $29.9\pm3.2$ |
> | Linear Decay | $6.8\pm5.2$ | $16.8\pm4.4$ | $27.0\pm3.8$ | $29.2\pm3.7$ | $31.2\pm4.8$ | $28.8\pm3.1$ | $33.6\pm3.4$ | $35.3\pm2.5$ | $35.1\pm2.7$ | $30.7\pm7.6$ | $32.9\pm5.8$ |
>
>
> ### Q2
> The purpose of \\(L_{\text{energy}}\\) is not to expose the teacher to future tasks, but to enforce a separation boundary between states that are familiar to the teacher and states that have never appeared in its training distribution. In the main paper, we use random rollout states in the student envrionment as OOD examples simply because they provide a concrete and intuitive illustration of this boundary.
>
> To address the concern about future leakage, we conduct a new experiment in the *Unlocked-to-Locked* setting where OOD states are sampled uniformly from the full MiniGrid observation space (10 seeds). This sampling does not assume anything about the target task; it draws from all valid observations that could occur in the domain, and we match the sample size to the original setup for a fair comparison. This removes any possibility of information leakage. The results below, along with the visualizations in Appendix G, show that the method behaves consistently under this alternative and continues to produce strong transfer improvements. ($q=-1$ is the Action Advising Baseline).
>
> |Decay|-1.0|0.0|0.1|0.2|0.3|0.4|0.5|0.6|0.7|0.8|0.9|
> |----|----|----|----|----|----|----|----|----|----|----|----|
> |25%|$20.6\pm8.3$|$29.0\pm5.9$|$35.6\pm7.9$|$42.6\pm2.8$|$42.4\pm3.9$|$35.1\pm7.6$|$39.4\pm4.2$|$39.7\pm10.7$|$39.9\pm5.8$|$38.0\pm3.8$|$24.3\pm7.6$|
> |50%|$14.6\pm3.0$|$19.2\pm5.2$|$30.7\pm3.3$|$31.2\pm2.3$|$29.5\pm3.9$|$34.8\pm3.2$|$36.3\pm2.7$|$38.2\pm2.9$|$39.4\pm2.2$|$34.1\pm3.9$|$33.6\pm3.2$|
> |75%|$-2.0\pm3.0$|$3.4\pm2.0$|$8.2\pm3.5$|$13.6\pm3.6$|$18.8\pm4.1$|$23.4\pm4.7$|$25.8\pm3.9$|$31.9\pm4.0$|$34.6\pm4.4$|$30.7\pm3.1$|$35.3\pm3.0$|
>
>
>
> ### Q3
> The newly added Appendix.H reports both the rate at which the teacher issues advice and the rate at which the student actually follows that advice (accounting for the decay), across multiple choices of \\( q \\) and at different points in training under various decay schedules. As training progresses, the issue-advice rate increases rather than decreases: once the student begins to master the task, it visits states that fall within the teacher’s training distribution more frequently, which causes the teacher to classify a larger portion of states as familiar. The take-advice rate is simply the issue-advice rate multiplied by the linear decay schedule, so it naturally decreases over time. These measurements characterize how the teacher’s influence changes throughout training and demonstrate that EBTL provides a controlled transition from teacher-guided to student-driven behavior.

---

### Comment · Area_Chair_FSsz · 2025-11-21
**Author-Reviewer Discussion**

Dear reviewers,

Please review the authors' response and adjust your rating accordingly. If you have any further questions, please discuss with the authors further.

AC

---

### Meta-Review · Area_Chair_wCWX · 2026-01-06

**Summary:**

This paper proposes Energy-Based Transfer Learning (EBTL), a selective teacher–student transfer framework in which a pretrained teacher provides action advice only in states deemed in-distribution according to an energy-based familiarity score. Reviewers generally agreed that negative transfer is an important problem and that the proposed mechanism is simple and empirically effective in the presented benchmarks, especially after the authors’ substantial rebuttal additions (continuous-control experiments, ablations disentangling energy gating from decay, and robustness analyses). However, a fundamental concern remains regarding the underlying problem formulation and assumptions. In particular, the method relies on a strong and largely artificial assumption that a teacher—who does not know the target reward—nonetheless takes reward-aligned optimal actions whenever the current state lies within its training distribution. This assumption holds in the paper’s carefully constructed environments, where task identity and reward-relevant structure are explicitly encoded in the state and optimal behavior decomposes cleanly into reusable subtasks, but it is difficult to reconcile with realistic RL settings where optimal actions depend on future objectives, long-horizon context, or task-specific intent rather than state familiarity alone. As a result, while the energy-gating mechanism is coherent and well-evaluated within the paper’s domains, the broader applicability of the approach remains unclear. Overall, the paper makes a solid methodological contribution and provides useful empirical insights into mitigating negative transfer under distribution shift, but its practical relevance is constrained by a teacher model that is unlikely to exist outside highly structured or synthetic environments.

**Reviewer Concerns:**

The concerns about experiments were addressed but the ones about motivation are not.

**Reviewer Scores:**

NA

---

### Decision · Program_Chairs · 2026-01-26

Reject